THE
EMBO
JOURNAL

# Ribonuclease H2 mutations induce a cGAS/STING-dependent innate immune response

Karen J Mackenzie[1], Paula Carroll[1], Laura Lettice[1], Žygimantė Tarnauskaitė[1], Kaalak Reddy[1], Flora Dix[1], Ailsa Revuelta[1], Erika Abbondati[2], Rachel E Rigby[1,†], Björn Rabe[1,‡], Fiona Kilanowski[1], Graeme Grimes[1], Adeline Fluteau[1], Paul S Devenney[1], Robert E Hill[1], Martin AM Reijns[1] & Andrew P Jackson[1,*]

## Abstract

Aicardi–Goutières syndrome (AGS) provides a monogenic model of nucleic acid-mediated inflammation relevant to the pathogenesis of systemic autoimmunity. Mutations that impair ribonuclease (RNase) H2 enzyme function are the most frequent cause of this autoinflammatory disorder of childhood and are also associated with systemic lupus erythematosus. Reduced processing of either RNA:DNA hybrid or genome-embedded ribonucleotide substrates is thought to lead to activation of a yet undefined nucleic acid-sensing pathway. Here, we establish $Rnaseh2b^{A174T/A174T}$ knock-in mice as a subclinical model of disease, identifying significant interferon-stimulated gene (ISG) transcript upregulation that recapitulates the ISG signature seen in AGS patients. The inflammatory response is dependent on the nucleic acid sensor cyclic GMP-AMP synthase (cGAS) and its adaptor STING and is associated with reduced cellular ribonucleotide excision repair activity and increased DNA damage. This suggests that cGAS/STING is a key nucleic acid-sensing pathway relevant to AGS, providing additional insight into disease pathogenesis relevant to the development of therapeutics for this childhood-onset interferonopathy and adult systemic autoimmune disorders.

**Keywords** Aicardi–Goutières syndrome; autoinflammation; cGAS-STING; ribonuclease H2

**Subject Categories** Immunology

The EMBO Journal (2016) 35: 831–844

See also: **M Gentili & N Manel** (April 2016)

## Introduction

Nucleic acid sensing is a key element of antimicrobial immunity (Wu & Chen, 2014). Diverse nucleic acid sensors act as pattern recognition receptors (PRRs), initiating innate immune responses

necessary to protect against pathogens containing a diverse range of nucleic acid species (Wu & Chen, 2014). Such innate immune activation, arising from sensing of cellular nucleic acids, is also implicated in autoimmune disease, most notably systemic lupus erythematosus (SLE) (Wahren-Herlenius & Dorner, 2013).

The autoinflammatory disorder Aicardi–Goutières syndrome (AGS) has provided important insights into mechanisms underlying nucleic acid-mediated inflammation (Crow & Manel, 2015). AGS typically presents in infancy following a period of normal development, with sterile pyrexia, irritability, seizures and loss of developmental milestones (Crow & Livingston, 2008). Persisting severe physical and intellectual disability is frequent, and extra-neurological features can include vasculitic skin lesions (Crow & Livingston, 2008; Crow *et al*, 2015). Increased type 1 interferon activity is most reliably detected during the early stages of the disease (Crow *et al*, 2015). However, upregulated interferon-stimulated gene (ISG) transcripts in blood have proven to be the most robust diagnostic biomarker in AGS patients, and an "ISG signature" often persists long after the initial stages of disease (Rice *et al*, 2013; Crow *et al*, 2015).

Although AGS is a monogenic disorder, it is genetically heterogeneous, with seven genes implicated to date, encoding several nucleic acid processing enzymes and a cytosolic nucleic acid sensor. These comprise the RNASEH2A, RNASEH2B and RNASEH2C proteins of the RNase H2 endonuclease complex (Crow *et al*, 2006b) as well as TREX1, SAMHD1, ADAR and IFIH1 (Crow *et al*, 2006a; Rice *et al*, 2009, 2012, 2014). Heterozygous mutations in the three RNase H2 genes (Günther *et al*, 2015) and *TREX1* (Lee-Kirsch *et al*, 2007) are also associated with systemic lupus erythematosus.

Partial loss-of-function biallelic mutations in the RNase H2 genes are the major cause of AGS, accounting for over half of all cases (Crow *et al*, 2015). RNase H2 is ubiquitously expressed and functions alongside RNase H1 to degrade cellular RNA:DNA heteroduplexes. Although the exact *in vivo* substrates of these nucleases remain to be identified, they are thought to act on RNA:DNA hybrids such as those arising during nuclear DNA replication and R-loop formation (Cerritelli & Crouch, 2009). Unlike RNase H1, RNase H2 also cleaves and initiates the removal of single ribonucleotides

1  MRC Human Genetics Unit, MRC Institute of Genetics and Molecular Medicine, The University of Edinburgh, Edinburgh, UK
2  Roslin Institute, The University of Edinburgh, Edinburgh, UK
   *Corresponding author. Tel: +44 131 651 8500; Fax: +44 131 467 8456; E-mail: andrew.jackson@igmm.ed.ac.uk
   †Present address: MRC Human Immunology Unit, Radcliffe Department of Medicine, MRC WIMM University of Oxford, Oxford, UK
   ‡Present address: Institute of Biochemistry, Christian-Albrechts-University of Kiel, Kiel, Germany

embedded in DNA (Eder *et al*, 1993; Rydberg & Game, 2002), a process known as ribonucleotide excision repair (RER) (Sparks *et al*, 2012). RNase H2 is essential for mammalian genome stability, with complete loss of RNase H2 resulting in embryonic lethality (Reijns *et al*, 2012; Hiller *et al*, 2012).

Intracellular accumulation of aberrant nucleic acid species is believed to trigger intrinsic nucleic acid sensors initiating auto-immunity in AGS (Crow & Manel, 2015). $Trex1^{-/-}$ and $Adar1^{-/-}$ mouse models have, respectively, implicated the cGAS-STING (Ablasser *et al*, 2014; Ahn *et al*, 2014a; Gray *et al*, 2015; Gao *et al*, 2015) and MDA5/IFIH1 (Liddicoat *et al*, 2015) pathways in driving immune activation. However, to date, an immune phenotype in mice has not been described for mutations in any of the RNase H2 genes. It therefore remains to be determined whether a similar pathophysiological mechanism underlies RNase H2 deficiency, and consequently, the nature of the activated immune pathway in the majority of AGS patients is yet to be defined.

Here, we identify a sub-clinical phenotype of ISG induction in a mouse model of the *RNASEH2B-A177T* mutation, the single most common missense mutation found in AGS patients. We demonstrate that this is dependent on the cGAS/STING pathway, consistent with PRR sensing of cell-intrinsic nucleic acids in RNase H2-deficient cells.

# Results

### A hypomorphic RNase H2 mouse model for Aicardi–Goutières syndrome

A mouse model was created by targeted knock-in of the A174T missense mutation (c.520G>A) into exon 7 of *Rnaseh2b* in mouse embryonic stem cells using homologous recombination (Fig 1A–D and Appendix Fig S1). A C57BL/6J congenic $Rnaseh2b^{A174T/A174T}$ mouse line was established using these cells, orthologous to the most common pathogenic mutation identified in patients with AGS, *RNASEH2B-A177T*. This mutation resulted in reduced cellular levels of all three RNase H2 subunits in $Rnaseh2b^{A174T/A174T}$ mouse embryonic fibroblasts (MEFs) and $RNASEH2B^{A177T/A177T}$ AGS patient lymphoblastoid cells (LCLs) (Fig 1E). This is consistent with reduced RNase H2 complex stability predicted from structural and biochemical studies that showed that the RNASEH2B-RNASEH2C interaction interface is disrupted by the A177T substitution (Figiel *et al*, 2011; Reijns *et al*, 2011). Cellular RNase H2 activity was also significantly reduced to $30 \pm 2\%$ activity in $Rnaseh2b^{A174T/A174T}$ MEFs (Fig 1F and G) and $49 \pm 3\%$ activity in $RNASEH2B^{A177T/A177T}$ LCLs (Fig 1H), assessed against an embedded ribonucleotide substrate. More pronounced reduction in the mouse cells may be explained by the presence of a neomycin selection cassette between exon 6 and 7, causing reduced *Rnaseh2b* transcript levels (~60% of wild type; data not shown).

Despite marked impairment of RNase H2 activity, $Rnaseh2b^{A174T/A174T}$ mice had no overt phenotype and remained healthy when aged. Full pathological examination of brain, liver, heart, lungs, thymus, spleen, gastrointestinal tract, kidneys, skin and tongue from mice ($n = 9$) did not detect histological features of inflammation, infection or neoplasia at 1 year (data not shown). In particular, there was no evidence of intracranial calcification,

leukodystrophy, chilblain vasculitis or cardiomyopathy, clinical features associated with AGS in humans (Crow *et al*, 2015). Notably, histopathological signs of inflammation are also not evident in $Samhd1^{-/-}$ mice, although activation of innate immune signalling does occur, with ISG upregulation evident (Behrendt *et al*, 2013; Rehwinkel *et al*, 2013). We therefore investigated whether this was also the case in $Rnaseh2b^{A174T/A174T}$ mice.

### ISG upregulation is present in tissues from $Rnaseh2b^{A174T/A174T}$ mice

Since an ISG transcriptional response is the most robust biomarker of inflammation in human patients (Rice *et al*, 2013), we performed RT–qPCR for a panel of AGS-relevant ISGs on RNA extracted from adult $Rnaseh2b^{A174T/A174T}$ mouse tissues. A broad upregulation of ISGs was detected in heart (Fig 2A) along with significant induction of a subset of ISGs in the kidney (Fig 2B), but without any ISG response evident in brain (Fig 2C). The twofold to fourfold induction of ISGs we observed in $Rnaseh2b^{A174T/A174T}$ heart tissue is comparable to the fourfold to sevenfold induction seen in $Samhd1^{-/-}$ mouse tissue (Rehwinkel *et al*, 2013). While an overt inflammatory phenotype is seen in $Trex1^{-/-}$ mice (Morita *et al*, 2004; Gall *et al*, 2012), pathological signs of neuroinflammation are not evident, although ISG upregulation in brain tissue can be detected (Pereira-Lopes *et al*, 2013). Given that the autoinflammatory process appears to initiate in the $Trex1^{-/-}$ heart (Gall *et al*, 2012), there are similarities between the pattern of inflammation in $Trex1^{-/-}$ mice and the ISG tissue expression pattern in $Rnaseh2b^{A174T/A174T}$ mice.

### A proinflammatory response in $Rnaseh2b^{-/-}$ MEFs

We next investigated whether the observed ISG induction was directly attributable to reduced RNase H2 activity. To address this, we examined cells from a second independent mouse line carrying a null allele of *Rnaseh2b* ($Rnaseh2b^{tm1d}$), derived from the EUCOMM knockout-first $Rnaseh2b^{tm1a}$ allele, from which exon 5 of *Rnaseh2b* had been excised by Cre recombinase. Given the early embryonic lethality of $Rnaseh2b^{-/-}$ mice (Reijns *et al*, 2012), mouse embryonic fibroblasts (MEFs) were generated on a $p53^{-/-}$ background. Here-after, we will refer to the resulting $Rnaseh2b^{-/-}$ $p53^{-/-}$ and $Rnaseh2b^{+/+}$ $p53^{-/-}$ cells simply as $Rnaseh2b^{-/-}$ and $Rnaseh2b^{+/+}$, respectively. Loss of RNase H2 activity in $Rnaseh2b^{-/-}$ cells was confirmed (Fig 3A).

Significant upregulation of ISG transcripts was again apparent, on comparing six independent $Rnaseh2b^{-/-}$ MEF lines with four $Rnaseh2b^{+/+}$ control MEF lines (Fig 3B and Appendix Fig S2A). To further characterise the nature of the response, two of the $Rnaseh2b^{-/-}$ MEF lines with stronger ISG upregulation were compared to control lines by gene expression microarray (Fig 3C and Appendix Table S1). Of the 29 upregulated genes achieving genome-wide significance, 17 were documented ISGs and activation of non-ISG immune response pathways was not evident. In particular, there was significant induction of two ISG genes encoding proinflammatory cytokines: *Ccl5* ($P = 0.0067$) and *Cxcl10* ($P = 0.039$). Upregulation of both transcripts was confirmed by RT–qPCR, and ELISA of culture supernatants from all six $Rnaseh2b^{-/-}$ lines demonstrated significantly increased CCL5 and CXCL10 secretion (Fig 3D and E,

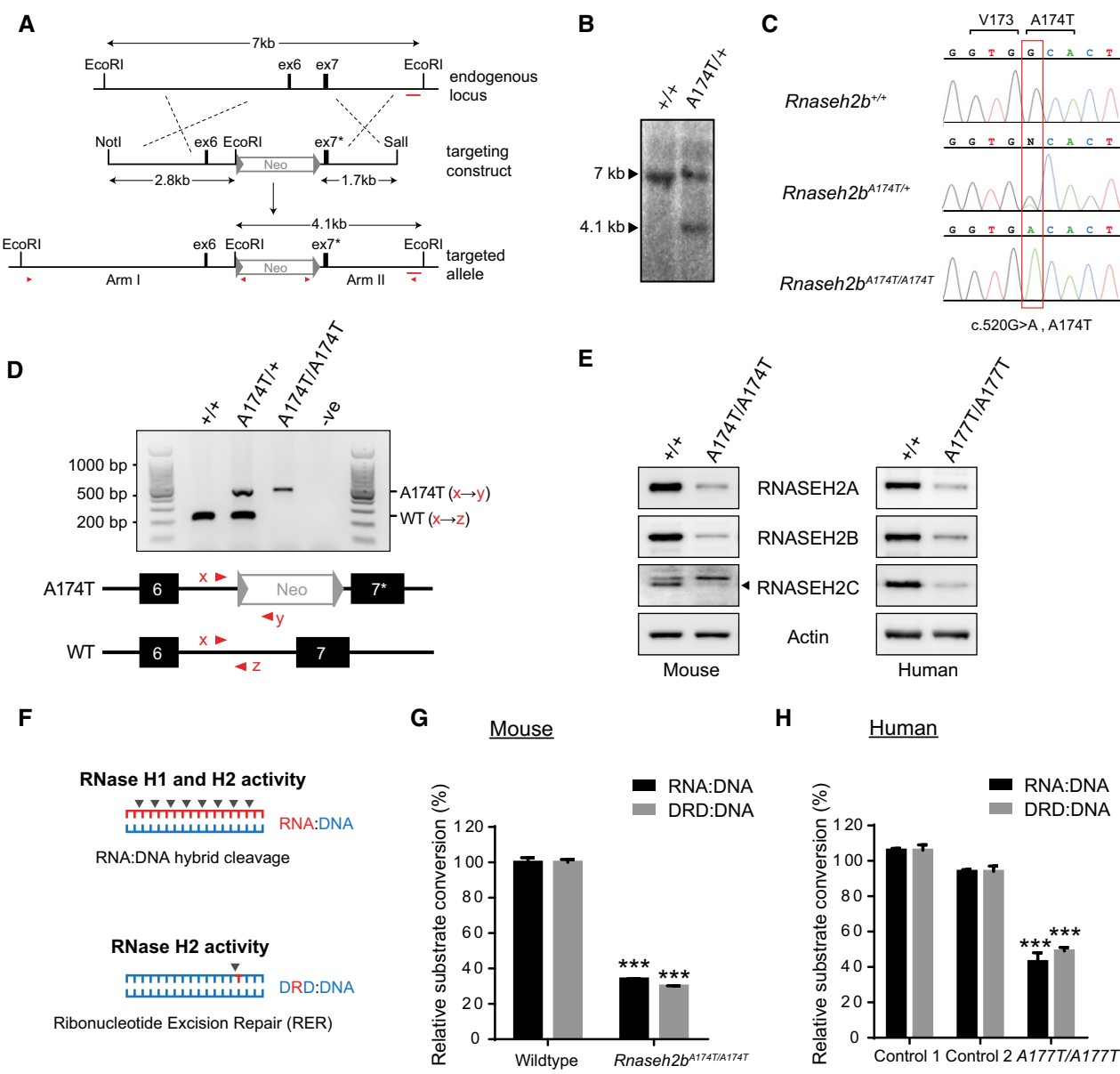

**Figure 1.   RNase H2 complex levels and enzymatic activity are reduced in *Rnaseh2b*<sup>A174T/A174T</sup> mouse and *RNASEH2B*<sup>A177T/A177T</sup> AGS patient cells.**

A       Targeted mutagenesis of the *Rnaseh2b* gene. Top: A 7-kb region of the *Rnaseh2b* genomic locus; black boxes, exons 6 (ex6) and 7 (ex7). Middle: NotI/SalI restriction fragment of the targeting construct, comprising 4.5 kb of genomic DNA and a neomycin selection cassette (Neo) flanked by loxP sites (triangles). (Bottom) Targeted locus containing exon 7 with the c.520G>A mutation (ex7*). Red arrowheads, primers used to confirm correct targeting. Red bar, 400-bp probe for Southern blotting.

B       Southern blotting confirms successful targeting. Introduction of an additional EcoRI site results in a 4.1-kb restriction fragment detectable by Southern for targeted ES cells (A174T/+) but not for parental DNA (+/+).

C       Capillary sequencing for *Rnaseh2b*<sup>+/+</sup>, *Rnaseh2b*<sup>A174T/+</sup> and *Rnaseh2b*<sup>A174T/A174T</sup> DNA confirmed the presence of the introduced missense mutation.

D       Mouse genotyping by multiplex PCR. Top: A 221-bp PCR product is present in wild-type mice (+/+); the *Rnaseh2b*<sup>A174T</sup> allele (also) gives a 460-bp product. Bottom: Position of forward (x) and reverse primers (y, z).

E       Immunoblotting demonstrates depletion of all three RNase H2 protein subunits in *Rnaseh2b*<sup>A174T/A174T</sup> MEFs and *RNASEH2B*<sup>A177T/A177T</sup> LCLs. Representative of three independent experiments.

F       Schematic showing enzyme activities attributed to RNase H1 and RNase H2 (DNA blue, RNA red).

G, H   RNase H2 enzyme activity is reduced in mouse and patient cells. (G) Enzyme activity for *Rnaseh2b*<sup>A174T/A174T</sup> MEFs and passage-matched *Rnaseh2b*<sup>+/+</sup> controls, against RNase H substrate (RNA:DNA heteroduplex) and RNase H2-specific substrate, double-stranded DNA with a single-embedded ribonucleotide (DRD:DNA). Mean activity for three independent cell lines, error bars represent SEM. Enzymatic activity expressed relative to the average value of control MEFs. ***P < 0.001, two-tailed *t*-test (n = 3 *Rnaseh2b*<sup>A174T/A174T</sup> and n = 3 *Rnaseh2b*<sup>+/+</sup> control MEF lines). (H) RNase H2 activity in LCLs from two independent healthy controls and an AGS patient homozygous for the *RNASEH2B-A177T* mutation. Enzyme activity normalised to average activity of control lines. Three independent experiments, error bars represent SEM. ***P < 0.001 versus either control, two-tailed *t*-test.

Source data are available online for this figure.

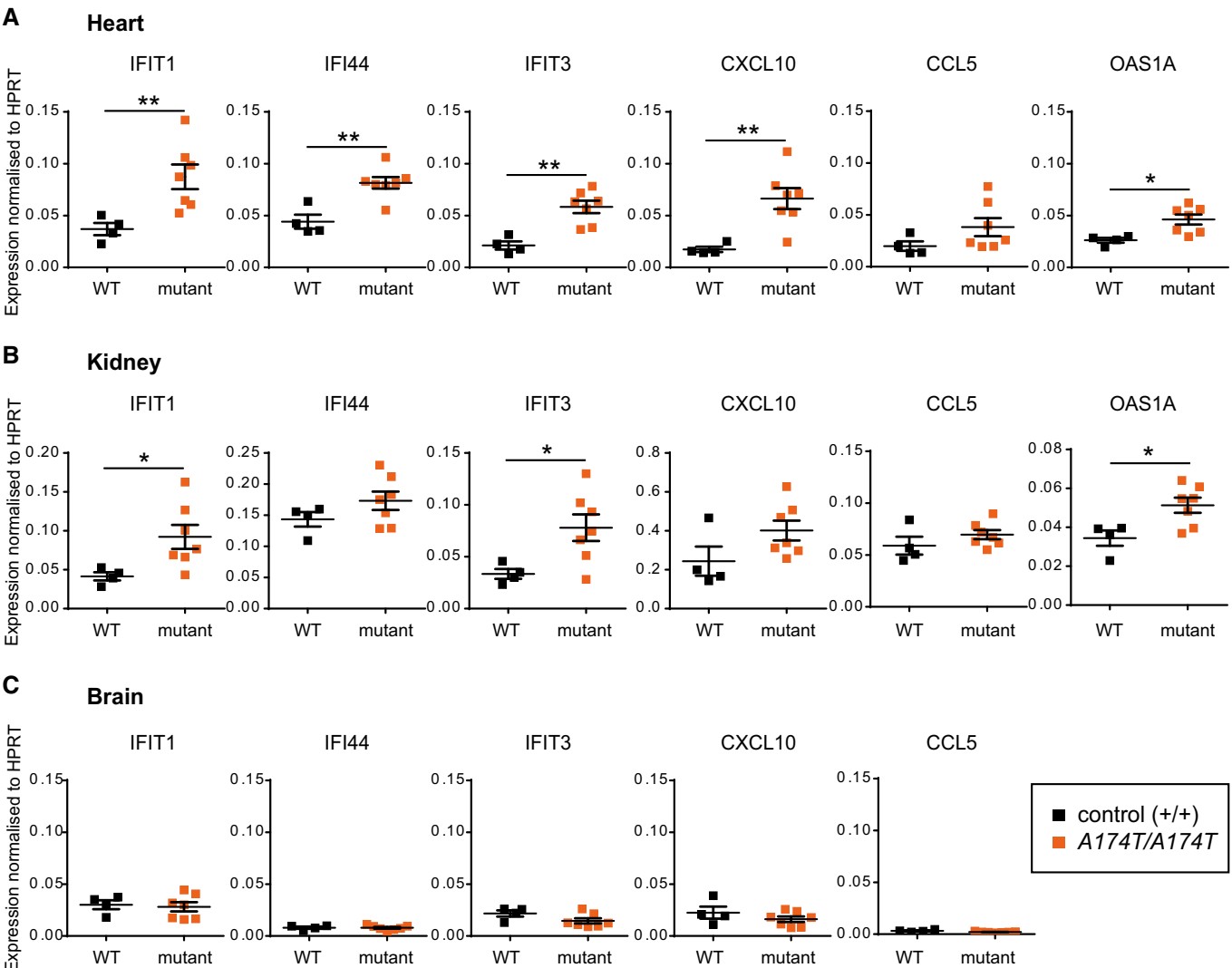

**Figure 2.  Increased ISG expression in tissues from *Rnaseh2b^A174T/A174T^* mice.**

A   Transcript levels of multiple ISGs are significantly elevated in heart.
B   Transcript levels of a subset of ISGs are significantly increased in kidney.
C   No ISG upregulation is evident in the brain.

Data information: ISG transcript levels determined by RT–qPCR normalised to transcript levels of the housekeeping gene HPRT (*Oas1a* was undetectable in brain). Each data point represents the mean of technical replicates of tissue RNA from a single mouse. $n = 9$ nine-month-old *Rnaseh2b^A174T/A174T^* mice and $n = 4$ age-matched control wild-type C57BL/6J mice. Horizontal line, mean; error bars, SEM. *$P < 0.05$, **$P < 0.01$, two-tailed *t*-test.

and Appendix Fig S2B). Notably, these cytokines have previously been implicated in AGS neuroinflammation in humans (van Heteren *et al*, 2008; Takanohashi *et al*, 2013; Cuadrado *et al*, 2015). We therefore concluded that impaired RNase H2 activity *in vitro* and *in vivo* results in a similar inflammatory response to that observed in patients with AGS, and like for *Trex1* deficiency (Gall *et al*, 2012) is present in non-immune cells.

### ISG activation is dependent on the cGAS-STING nucleic acid-sensing pathway

We next sought to determine the nucleic acid-sensing pathway responsible for ISG induction in *Rnaseh2b^−/−^* cells. Three

pathways have recently been implicated in the sensing of RNA: DNA hybrids: TLR9 (Rigby *et al*, 2014), cGAS-STING (Mankan *et al*, 2014) and the NLRP3 inflammasome (Kailasan Vanaja *et al*, 2014). We therefore postulated that one of these nucleic acid sensors would be responsible for the ISG signature, given that RNase H2 is the major enzyme degrading RNA:DNA hybrids in mammalian cells (Büsen, 1980; Reijns *et al*, 2012). However, as IL-1β and IL-18 were both undetectable in culture supernatants from *Rnaseh2b^−/−^* cells (data not shown), an inflammasome-mediated response was unlikely. While the observed ISG induction would be consistent with TLR9 activation, signalling is impaired by deficient proteolytic processing of the receptor in MEFs (Ewald *et al*, 2008), and we therefore prioritised assessment

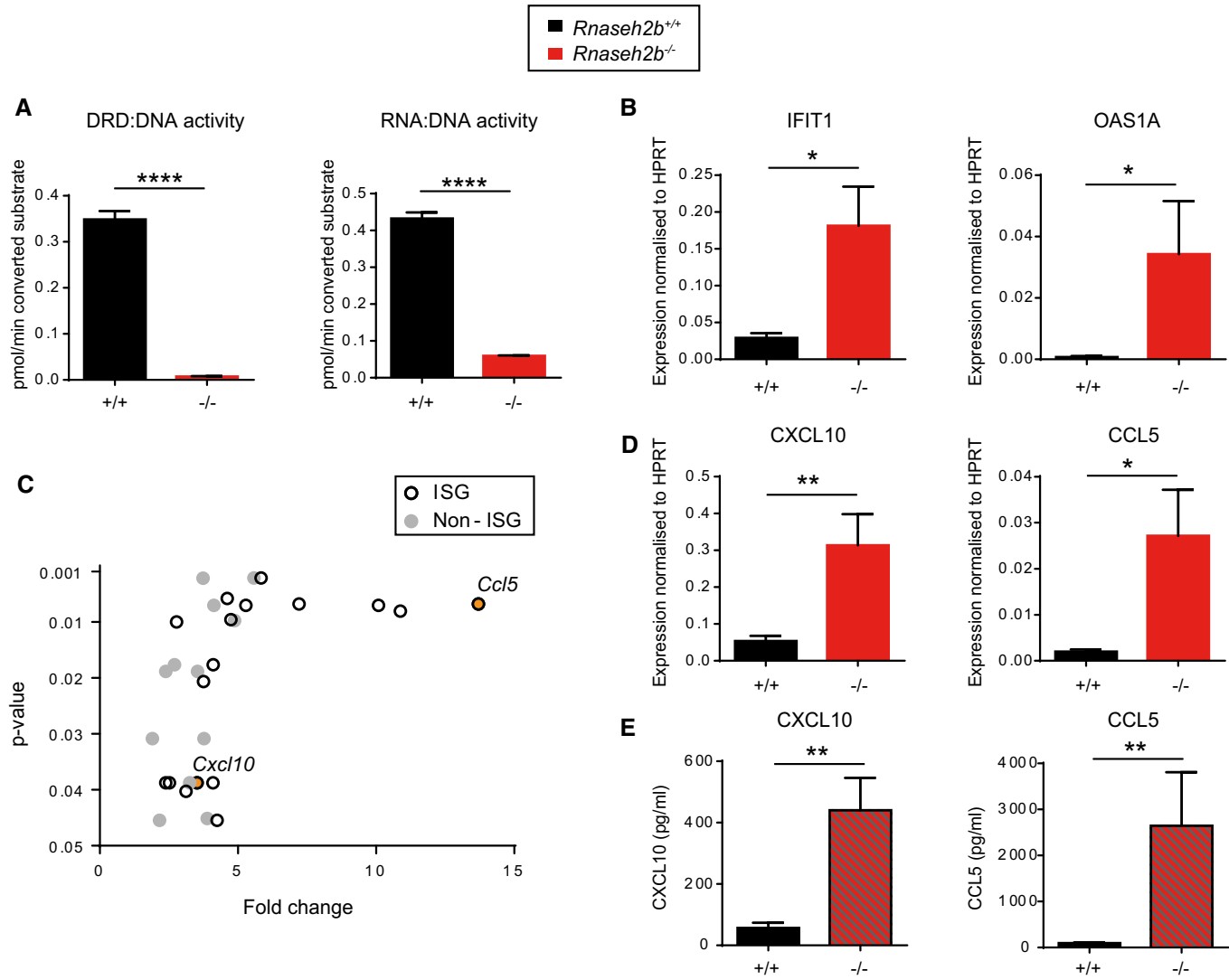

**Figure 3.  A proinflammatory response is detectable *in vitro* in *Rnaseh2b*<sup>−/−</sup> MEFs.**

*Rnaseh2b*<sup>−/−</sup> MEFs have significantly elevated ISG transcript levels and secrete the proinflammatory cytokines CXCL10 and CCL5.

A  Validation of *Rnaseh2b*<sup>−/−</sup> MEF lines. RER activity (DRD:DNA) is undetectable in a *Rnaseh2b*<sup>−/−</sup> line consistent with complete inactivation of the *Rnaseh2b* gene (Reijns *et al*, 2012). Mean of three independent experiments; error bars, SEM, ****$P < 0.0001$, two-tailed *t*-test. *Rnaseh2b*<sup>−/−</sup> and *Rnaseh2b*<sup>+/+</sup> control MEFs on a C57BL/6 *p53*<sup>−/−</sup> background.

B  ISG transcript levels are increased in *Rnaseh2b*<sup>−/−</sup> MEFs.

C  BeadArray transcript analysis (Illumina) detected induction of multiple ISGs, including the cytokines CXCL10 and CCL5, but not other cytokine transcripts in *Rnaseh2b*<sup>−/−</sup> MEFs. Plotted, average fold enrichment versus *P*-value of significantly upregulated transcripts ($P < 0.05$, after multiple testing correction) comparing two *Rnaseh2b*<sup>−/−</sup> MEF lines versus 4 *Rnaseh2b*<sup>+/+</sup> MEF lines. Seventeen out of 29 transcripts are ISGs.

D  *Cxcl10* and *Ccl5* transcripts (detected by RT–qPCR) are significantly elevated.

E  Increased CXCL10 and CCL5 protein (detected by ELISA) is secreted by *Rnaseh2b*<sup>−/−</sup> MEFs.

Data information: Data in (B, D, E) are mean from three experiments for six independent *Rnaseh2b*<sup>−/−</sup> MEF lines versus four independent *Rnaseh2b*<sup>+/+</sup> MEF lines. Error bars, SEM. *$P < 0.05$, **$P < 0.01$, Mann–Whitney *U*-test.

of the cGAS-STING pathway. We performed siRNA depletion of cGAS in *Rnaseh2b*<sup>−/−</sup> MEFs and found that cGAS depletion significantly abrogated the ISG response and CCL5 production (Fig 4A). siRNA depletion of STING, the adaptor associated with cGAS sensing, also significantly reduced ISG induction and CCL5 secretion (Fig 4A), implicating the cGAS/STING-sensing pathway in innate immune activation in *Rnaseh2b*<sup>−/−</sup> cells. The siRNA knock-down

was specific to cGAS and STING, respectively (Fig 4A), and cells remained fully responsive to poly(I:C) (Appendix Fig S3).

To substantiate that cGAS is required for ISG induction, CRISPR/Cas9 genome editing was performed to knock out the cGAS gene in *Rnaseh2b*<sup>−/−</sup> MEFs. *Rnaseh2b*<sup>−/−</sup> MEF clones in which cGAS deletion had not occurred were also identified, to act as experimental controls. While ISG transcript upregulation and cytokine production

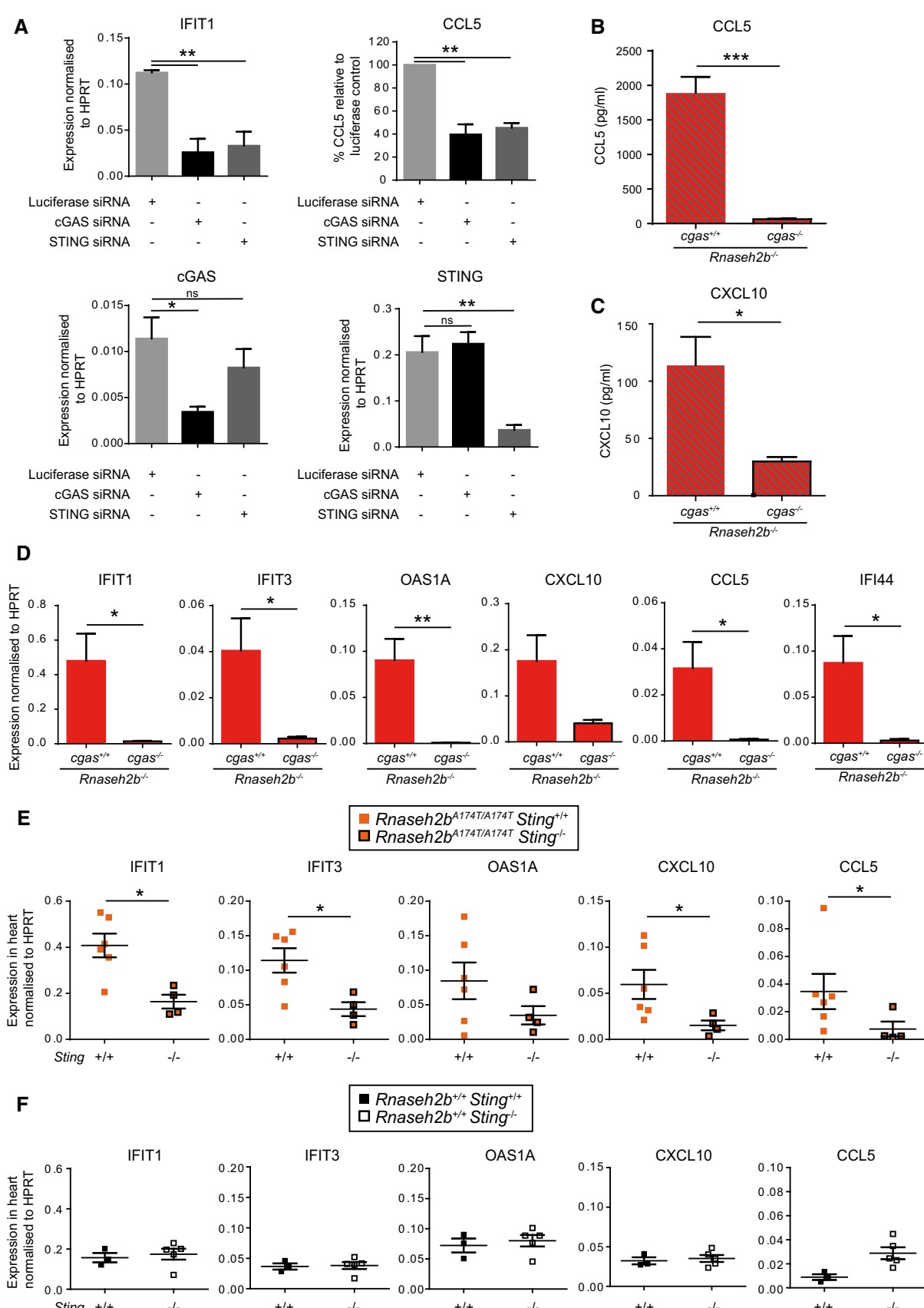

**Figure 4.**

**Figure 4. ISG induction in RNase H2 deficiency is dependent on the cGAS-STING nucleic acid-sensing pathway.**

A    ISG activation and cytokine secretion in $Rnaseh2b^{-/-}$ MEFs is markedly impaired by cGAS or STING siRNA depletion. Upper left and lower panels: RT–qPCR of *Ifit1*, *cGas* and *Sting* transcripts, 48 h after siRNA targeting luciferase (control), cGAS or STING. Upper right panel: CCL5 is significantly reduced in culture supernatants 48 h after cGAS or STING depletion. Concentration of CCL5 (ELISA), normalised to luciferase siRNA control levels in each experiment. Mean from three independent experiments using one $Rnaseh2b^{-/-}$ MEF line; error bars, SEM. \*$P < 0.05$, \*\*$P < 0.01$, two-tailed *t*-test for RT–qPCR, one-sample *t*-test for CCL5 ELISA.

B–D  ISG induction and cytokine secretion is abolished in $Rnaseh2b^{-/-}$ $cGas^{-/-}$ MEFs. *cGas* was targeted by CRISPR/Cas9 genome editing of a $Rnaseh2b^{-/-}$ MEF line to inactivate cGAS/STING signalling. In addition to sequence validation, functional inactivation of cGAS was confirmed in $Rnaseh2b^{-/-}$ $cGas^{-/-}$ CRISPR lines by the absence of CCL5 secretion in response to dsDNA (Appendix Fig S4). CCL5 (B) and CXCL10 (C) production, as well as ISG expression (D), was abrogated in $Rnaseh2b^{-/-}$ $cGas^{-/-}$ clones, assessed by ELISA and RT–qPCR, respectively. Four independent experiments, $n = 2$ $Rnaseh2b^{-/-}$ $cGas^{-/-}$ clones, $n = 4$ $Rnaseh2b^{-/-}$ $cGas^{+/+}$ clones, error bars, SEM of each experiment; \*$P < 0.05$, \*\*$P < 0.01$, \*\*\*$P < 0.001$, two-tailed t-test.

E    ISG induction in $Rnaseh2b^{A174T/A174T}$ mice is STING dependent. RT–qPCR of RNA extracted from hearts from $Sting^{+/+}$ $Rnaseh2b^{A174T/A174T}$ ($n = 4$) and $Sting^{-/-}$ $Rnaseh2b^{A174T/A174T}$ ($n = 6$) 3-month-old mice. Each data point represents the mean of technical triplicates from one mouse. Horizontal line, mean; error bars, SEM. \*$P < 0.05$, Mann–Whitney *U*-test.

F    Absence of STING does not significantly decrease basal ISG expression. RT–qPCR of RNA extracted from hearts from $Sting^{+/+}$ ($n = 3$) and $Sting^{-/-}$ ($n = 5$) three-month-old mice. Each data point represents the mean of technical triplicates from one mouse. Horizontal line, mean; error bars, SEM. \*$P < 0.05$, Mann–Whitney *U*-test.

(CCL5 and CXCL10) were still observed in such control cells, both were abrogated in CRISPR/Cas9 targeted $Rnaseh2b^{-/-}$ $cGas^{-/-}$ cells (Fig 4B–D), establishing that the innate immune activation found in $Rnaseh2b^{-/-}$ MEFs is dependent on cGAS.

To determine whether ISG induction was dependent on the cGAS-STING-sensing pathway *in vivo*, we intercrossed $Rnaseh2b^{A174T/A174T}$ and $Sting^{-/-}$ mice. We found that ISG transcript levels from $Rnaseh2b^{A174T/A174T}$ $Sting^{-/-}$ heart tissue were significantly reduced compared to $Rnaseh2b^{A174T/A174T}$ $Sting^{+/+}$ controls (Fig 4E), while no significant reduction was observed when comparing $Rnaseh2b^{+/+}$ $Sting^{-/-}$ and $Rnaseh2b^{+/+}$ $Sting^{+/+}$ controls (Fig 4F). Hence, a STING-dependent ISG response also occurs *in vivo* in $Rnaseh2b^{A174T/A174T}$ mice, implicating the cGAS-STING pathway in the ISG induction observed in RNase H2 AGS patients.

**Loss of RNase H2-specific activity results in ISG induction**

Finally, the fact that both dsDNA (Sun *et al*, 2013; Gao *et al*, 2013) and RNA:DNA hybrids (Mankan *et al*, 2014) can bind and activate the cytoplasmic nucleic acid sensor cGAS prompted us to consider the origin of the cGAS ligand present in RNase H2-deficient cells. Abrogated RNase H2 function could give rise to cytosolic accumulation of RNA:DNA heteroduplexes as a consequence of reduced RNA:DNA degradation of structures such as R-loops or active retro-elements/endogenous retroviruses (Chon *et al*, 2013; Rigby *et al*, 2014; Moelling & Broecker, 2015). Alternatively, cytoplasmic DNA with embedded ribonucleotides may accumulate as a consequence of impaired RER that resulted in genome instability (Reijns *et al*, 2012; Hiller *et al*, 2012). To address these possibilities, we performed complementation experiments in RNase H2 null cells, to

establish which enzymatic activity was associated with the proinflammatory response.

$Rnaseh2b^{-/-}$ cells were complemented by retroviral transduction of either *Rnaseh1* or *Rnaseh2b,* respectively, to reconstitute cellular RNase H activity or RNase H plus RER activity. Overexpression of RNase H1 in $Rnaseh2b^{-/-}$ cells restored cellular enzyme activity against RNA:DNA hybrids to $81 \pm 10\%$ of the level seen in $Rnaseh2b^{+/+}$ cells (Fig 5A), but did not alleviate DNA damage (Fig 5B and C). Complementation with *Rnaseh2b* reconstituted activity against both types of substrates (Fig 5A) and also returned DNA damage to levels seen in $Rnaseh2b^{+/+}$ cells (Fig 5B and C). Reconstitution with *Rnaseh2b* was able to reduce cytokine and ISG responses close to wild-type levels (Fig 5D–F), while *Rnaseh1* complementation did not (Fig 5D–F). While some reduction was seen in CXCL10, ISG expression in *Rnaseh1* complemented cells was otherwise the same, if not greater than, in parental $Rnaseh2b^{-/-}$ cells. cGAS-dependent ISG induction in $Rnaseh2b^{-/-}$ cells is therefore associated with DNA damage and loss of RNase H2-specific activity, rather than an overall reduction in cellular activity against RNA:DNA hybrids.

# Discussion

Here, we establish that RNase H2 deficiency leads to a proinflammatory response which is dependent upon the cGAS/STING pathway. The resulting ISG transcriptional response and induction of proinflammatory cytokines are consistent with cell-intrinsic innate immune activation. ISG activation varied between tissues, which may explain why such transcriptional changes were not reported

**Figure 5. Cellular RER and not enzyme activity against RNA:DNA hybrids correlates with DNA damage and proinflammatory response.**

A    Overexpression of RNase H1 in $Rnaseh2b^{-/-}$ cells restores RNase H activity against RNA:DNA hybrids to $81 \pm 10\%$ of wild-type levels, while overexpression of RNASEH2B restores cellular enzyme activity for cleavage of both RNA:DNA and DRD:DNA substrates (RER). $Rnaseh2b^{-/-}$ MEFs were complemented with *Rnaseh1* (+H1), *Rnaseh2b* (+H2B) or EGFP by retroviral infection. Mean of $n = 3$ independent experiments $\pm$ SEM.

B, C  DNA damage is reduced to wild-type levels by complementation with *Rnaseh2b* but not *Rnaseh1*, measured by 53BP1 foci formation in detergent-extracted fixed cells. (B) Representative images (scale bar, 10 μm). (C) At least 150 cells were counted for each cell line in three independent experiments. Mean $\pm$ SEM, \*\*\*\*$P < 0.0001$ two-tailed *t*-test.

D–F  CCL5 (D) and CXCL10 production (E), as well as ISG induction (F) in $Rnaseh2b^{-/-}$ MEFs are reduced close to wild-type levels ($Rnaseh2b^{+/+}$), by complementation with *Rnaseh2b* but not *Rnaseh1*. Mean of $n = 6$ independent experiments $\pm$ SEM for complemented cells; $n = 3$ independent experiments for $Rnaseh2b^{-/-}$ parental and $Rnaseh2b^{+/+}$ controls cells. \*$P < 0.05$, \*\*$P < 0.01$, \*\*\*$P < 0.001$, \*\*\*\*$P < 0.0001$ two-tailed *t*-test indicates significantly reduced expression compared to $Rnaseh2b^{-/-}$ parental cells.

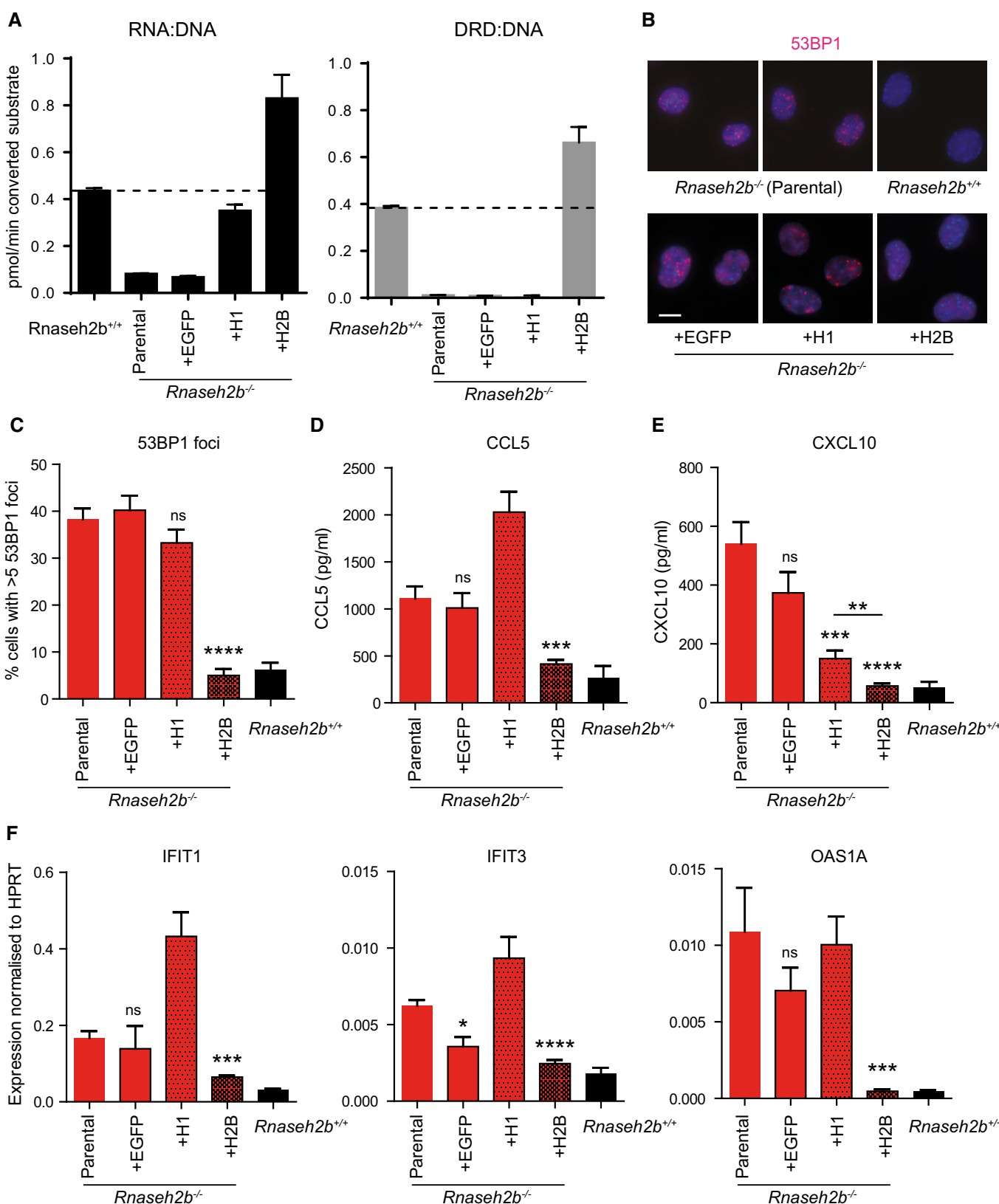

**Figure 5.**

previously in $Rnaseh2b^{-/-}$ or $Rnaseh2b^{tm1a/tm1a}$ embryos (Reijns *et al*, 2012; Hiller *et al*, 2012). The precise factors determining tissue-specific cGAS activation are currently unclear, but could include the sensitivity of the cGAS/STING pathway in particular cell types, their level of cell proliferation, the rate of accumulation of immunogenic nucleic acids and the counteracting influences of cellular processes degrading such nucleic acids.

Despite tissue-specific ISG upregulation, $Rnaseh2b^{A174T/A174T}$ mice have a subclinical phenotype without any overt inflammatory pathology, and so, like all other AGS mouse models do not recapitulate the neuroinflammation seen in human AGS patients (Rabe, 2013; Behrendt & Roers, 2014). It will therefore be important to confirm that the innate immune pathways implicated in Trex1, Adar1 and now RNase H2 deficiency, through the use of mouse models, are also relevant to the human autoinflammatory phenotype in AGS patients in whom these genes are affected. $Samhd1^{-/-}$ mice, like $Rnaseh2b^{A174T/A174T}$ mice, display an ISG response in the absence of detectable pathology (Behrendt *et al*, 2013; Rehwinkel *et al*, 2013). In contrast a strong ISG response in *Adar1* null or editing-deficient mice is associated with embryonic lethality (Mannion *et al*, 2014; Liddicoat *et al*, 2015; Pestal *et al*, 2015), and with autoinflammatory cardiomyopathy and multi-tissue involvement in $Trex1^{-/-}$ mice (Morita *et al*, 2004; Stetson *et al*, 2008; Gall *et al*, 2012). The variation in severity between different AGS gene mouse models remains unexplained, although it may be meaningful that mutations in human *RNASEH2B* are associated with the least severe disease course, with AGS onset generally in infancy, in contrast to the prenatal/neonatal onset more commonly seen in *TREX1* patients (Crow *et al*, 2015). Also, additional triggers, such as viral infection, have been proposed to be relevant to the pathogenesis of AGS (Crow & Manel, 2015). Reports of marked phenotypic variability in sibling pairs with identical *RNASEH2B* or *RNASEH2C* mutations (Vogt *et al*, 2013; Tüngler *et al*, 2014) are consistent with the possibility of such environmental factors impacting on disease severity. The $Rnaseh2b^{A174T/A174T}$ mouse and other models provide an opportunity for future investigation of these aspects.

Identification of the precise source of the immunogenic nucleic acids responsible for the inflammatory phenotype remains central to providing further mechanistic insight. Dependence of the ISG response in RNase H2- and Trex1-deficient mouse cells on the cGAS/STING pathway suggests that accumulation of cytoplasmic DNA is common to both TREX1 and RNase H2 AGS, given that DNA is the canonical ligand for cGAS (Sun *et al*, 2013; Gao *et al*, 2013). Our complementation experiments would favour this possibility, as normalising cellular enzymatic activity against RNA:DNA hybrids does not rescue the proinflammatory response. However, immunogenic RNA:DNA hybrids cannot be completely excluded, given that some hybrids might be specifically degraded by the RNase H2 enzyme (Chon *et al*, 2013), although the mechanism for such differential substrate specificity for RNase H1 and H2 enzymes is currently unclear.

So far, we have been unable to ascertain whether cytoplasmic double-stranded nucleic acids accumulate in RNase H2-deficient cells. However, given that impaired RER causes reduced genome stability, DNA fragments resulting from DNA strand breaks are a potential origin of immunogenic cytoplasmic DNA (Ahn *et al*, 2014b; Hartlova *et al*, 2015; Shen *et al*, 2015), while an alternative

source could be reverse-transcribed retroelements, which are known to be activated upon DNA damage (Farkash & Luning Prak, 2006). Irrespective of the chemical nature of the ligand, our observation that cGAS plays a central role in the ISG response in RNase H2 deficiency provides further impetus for defining the source of the immunogenic nucleic acids. While cytoplasmic DNA was first detected in 2007 in $Trex1^{-/-}$ cells, even here the precise origin still remains elusive, with conflicting evidence supporting either endogenous retroviruses/retroelements (Stetson *et al*, 2008) or genome instability (Yang *et al*, 2007). Purification of the responsible immunostimulatory nucleic acids could provide the solution; however, the specific isolation of such low-abundant cytoplasmic nucleic acids remains a formidable technical challenge. This has precluded a definitive answer to date, but may be aided by implicating cGAS as the nucleic acid sensor binding immunogenic nucleic acids in both Trex1- and RNase H2-deficient cells, informing potential future biochemical strategies.

In summary, our findings implicate the cGAS-STING pathway in RNase H2 AGS and, together with the previously attributed role in Trex1 deficiency, suggest it is the most common signalling pathway driving inflammation in AGS, relevant also to the pathogenesis of SLE (Lee-Kirsch *et al*, 2007; Günther *et al*, 2015). Targeting this pathway therefore represents a relevant therapeutic strategy for the treatment of this childhood interferonopathy and related adult-onset autoimmune conditions.

## Materials and Methods

### Generation of *Rnaseh2b-A174T* ES cells

Gap repair was used to retrieve a genomic fragment of 4.5 kb from BAC bMQ454F14 (Source Bioscience Lifesciences), which included exon 6 and 7 of the mouse *Rnaseh2b* locus (nucleotides 62977197-62981727 of Chr14, Ensembl release 64). Subsequent bacterial recombination was used to insert a neomycin cassette flanked by loxP sites between exon 6 and 7, generating a targeting vector with two external homology arms of 2.8 kb and 1.7 kb. A point mutation c.520G>A (A174T) was inserted into exon 7 of mouse *Rnaseh2b* by site-directed-mutagenesis (Quikchange, Agilent Technologies). After linearisation and vector backbone removal by digestion with NotI and SalI, and electroelution/purification by Elutrap, the targeting cassette was electroporated into 129/Ola E14Tg2AIV embryonic stem cells. Southern blotting and long-range PCR were used to identify correctly targeted clones. Capillary sequencing was performed to ensure the presence of the c.520G>A mutation and absence of other coding changes. The ES cell clone was karyotyped before injection into C57BL/6J mouse blastocysts.

### Mice

Rnaseh2b$^{A174T}$

Male chimeras resulting from blastocyst injection with $Rnaseh2b^{A174T/+}$ ES cells were crossed to C57BL/6J females, giving rise to heterozygous *Rnaseh2b* knock-in mice carrying the A174T mutation ($Rnaseh2b^{tm2-hgu-A174T}$, elsewhere referred to as $Rnaseh2b^{A174T}$). These were backcrossed to F11 on the C57BL/6J

background to establish congenicity and subsequently maintained as a homozygous mutant line. C57Bl/6J control mice were bought in at 4–6 weeks of age from the same source as those used in backcrossing, to ensure genetic matching to a level of > 99.97% identity. All mice were housed in the same facility in the same room in conventional cages and fed the same water and food until they were analysed. Murine data were analysed unblinded to genotype. Sample sizes of 3–7 mice per group were similar to previous studies.

### Rnaseh2b[tm1d]

Knockout-first *Rnaseh2b* mice were generated by blastocyst injection of the *Rnaseh2b[tm1a(EUCOMM)Wtsi]* C57BL/6N ES cell clone EPD0087_4_A02 (EUCOMM ID: 24441) and crossing of male chimeras to C57BL/6J females. The *Rnaseh2b[tm1d]* allele, in which exon 5 is deleted, was generated by crossing of *Rnaseh2b[tm1a/+]* mice to mice with ubiquitous expression of FLPe recombinase to delete the genetrap cassette. *Rnaseh2b[tm1c/+]* offspring were subsequently crossed to Cre745 mice (a kind gift from DJ Kleinjan, University of Edinburgh), containing a CAGGS-Cre construct in which Cre recombinase is under control of a chicken β-actin promoter (Araki *et al*, 1995; Kleinjan *et al*, 2006) to excise *Rnaseh2b* exon 5. The resulting *Rnaseh2b[tm1d/+]* (*Rnaseh2b[+/−]*) mice were maintained on the C57BL/6J background. The phenotype of *Rnaseh2b[tm1d/tm1d]* embryos was indistinguishable from that previously described for *Rnaseh2b[E202X/E202X]* embryos (Reijns *et al*, 2012).

### Other strains

*Trp53[tm1Tyj]*/J mice on the C57BL/6J background have been described previously (Jacks *et al*, 1994) and were kindly provided by Andrew Wood, University of Edinburgh. Interbreeding with *Rnasehb[tm1d/+]* mice was used to generate *Rnaseh2b[+/+] p53[−/−]* and *Rnaseh2b[−/−] p53[−/−]* MEFs.

*Sting/Mpys[−/−]* mice on the C57BL/6 background have been described previously (Jin *et al*, 2011) and were kindly provided by Jan Rehwinkel, University of Oxford, with the consent of John Cambier, University of Colorado SOM and National Jewish Health. Interbreeding with *Rnaseh2b[A174T/A174T]* mice generated *Rnaseh2b[A174T/A174T] Sting[−/−]* mice.

*Mb21d1/cGas[−/−]* MEFs derived from C57BL/6NTac-Mb21d1[tm1a(EUCOMM)Hmgu]/IcsOrl mice were a kind gift from Jan Rehwinkel, Oxford University. These mice were originally obtained from the Institute Clinique de la Souris through the European Mouse Mutant Archive.

All mouse studies were conducted according to UK Home Office regulations under a UK Home Office project licence.

### Genotyping

DNA was extracted from earclips (boiled in 50 μl 25 mM NaOH, 0.2 mM EDTA for 30 min at 95°C, cooled and then neutralised with 50 μl 40 mM Tris base) or embryos (tail tips treated with DirectPCR Lysis Reagent (Viagen) according to the manufacturer's instructions). All genotyping PCRs were performed using a multiplex three-primer strategy and Taq ReddyMix PCR Master Mix (Thermo Scientific), as previously described for *Rnaseh2b[E202X/E202X]* (Reijns *et al*, 2012) and *p53[−/−]* (Jacks *et al*, 1994). For *Rnaseh2b[tm1d/+]*,

*Sting/Mpys[+/−]* as well as all other primers and product sizes, see Appendix Table S2.

### Generation of murine embryonic fibroblast lines and assessment of innate immune activation

Independent *Rnaseh2b[A174T/A174T]* and *Rnaseh2b[+/+]* MEF lines were generated from individual E13.5 embryos. After removing the head and internal organs, the embryos were mechanically dissociated in growth medium (DMEM, 10% FCS, 50 U/ml penicillin and 50 μg/ml streptomycin, 0.1 mM β-mercaptoethanol). Resulting suspensions were grown at 37°C, 5% $CO_2$ and 3% $O_2$, and non-adherent cells removed after 24 h. MEFs were subsequently maintained and passaged under the same conditions. Independent *Rnaseh2b[−/−] p53[−/−]* and *Rnaseh2b[+/+] p53[−/−]* MEF lines were similarly generated from individual whole E10.5 embryos. For assessment of ISG upregulation and cytokine production, MEFs were plated at $2 \times 10^5$ cells per well in 12-well plates. The following day culture medium was replaced with 800 μl of fresh medium. After a further 20 h, the culture medium was removed for assessment by ELISA to determine CXCL10 and CCL5 concentrations (R&D Systems). RNA was extracted from adherent cells using the RNeasy kit (Qiagen) as per manufacturer's instructions and included DNase I treatment. Extracted RNA was stored at −80°C until analysis.

### AGS patient cells

A lymphoblastoid cell line from an AGS patient with the *RNASEH2B-A177T* mutation (kindly donated by Professor Yanick Crow, University of Manchester) and non-affected controls were generated from peripheral blood samples by EBV transformation using standard methods. Lymphoblastoid cell lines (LCLs) were maintained in RPMI 1640 supplemented with 15% foetal bovine serum, L-glutamine, 50 U/ml penicillin and 50 μg/ml streptomycin at 37°C, 5% $CO_2$ and normoxic conditions. The A177T/A177T mutation in the patient cell line was validated using Sanger sequencing.

### Immunoblotting

Whole-cell lysates were prepared by lysing cells in 50 mM Tris (pH 8), 280 mM NaCl, 0.5% NP-40, 0.2 mM EDTA, 0.2 mM EGTA, 10% glycerol (vol/vol), 1 mM DTT and 1 mM PMSF for 10 min at 4°C. Lysed cells were then diluted 1:1 with 20 mM HEPES (pH 7.9), 10 mM KCl, 1 mM EDTA, 10% glycerol (vol/vol), 1 mM DTT and 1 mM PMSF for an additional 10 min, and extracts were cleared by centrifugation (17,000 *g*, 5 min, 4°C). Equal amounts of proteins from supernatants (concentrations determined using the Bradford method) were separated by SDS–PAGE on NuPAGE Novex Bis-Tris 4–12% protein gels (Thermo Fisher Scientific) and transferred to PVDF membrane. Membranes were blocked in 5% milk in TBS with 0.2% Tween and probed with antibodies raised against mouse or human recombinant RNase H2, as previously described (Reijns *et al*, 2012), RNASEH2A (TA306706, Origene) or actin (A2066, Sigma).

### RNase H enzyme assays

Enzyme activity assays were performed using a FRET-based fluorescent substrate release assay, as previously described

(Reijns *et al*, 2011). Briefly, 10 mM of fluorescein-labelled oligonucleotides (GATCTGAGCCTGGGaGCT for RNase H2 specific activity, DRD:DNA, or gaucugagccugggagcu for total RNase H activity, RNA:DNA; upper case DNA, lower case RNA) was annealed to a complementary DABCYL-labelled DNA oligonucleotide (Eurogentec) in 60 mM KCl, 50 mM Tris–HCl pH 8. Activity against double-stranded DNA substrate of the same sequence was measured and used to correct for non-RNase H2 activity against DRD:DNA substrate. Reactions were performed in 100 μl of buffer (60 mM KCl, 50 mM Tris–HCl pH 8, 10 mM MgCl₂, 0.01% BSA, 0.01% Triton X-100) with 250 nM substrate in 96-well flat-bottomed plates at 24 ± 2°C. Whole-cell lysates were prepared as described above, and the final protein concentration used per reaction was 100 ng/μl (for DRD:DNA substrate), 50 ng/μl (for RNA:DNA substrate) for MEFs, and 32 or 16 ng/μl, respectively, for LCLs. Fluorescence was read for 100 ms using a VICTOR2 1420 multilabel counter (Perkin Elmer), with a 480-nm excitation filter and a 535-nm emission filter.

### RT–qPCR

Murine tissues were snap-frozen in liquid nitrogen and stored at −80°C until RNA extraction via homogenisation and using the RNeasy kit (Qiagen) as per manufacturer's instructions. cDNA was prepared from RNA from murine tissues or MEFs using AMV RT (Roche) and random oligomer primers (Thermo Fisher). Brilliant II SyBR Green qPCR Master Mix (Stratagene) was used to conduct RT–qPCR on the ABI Prism HT7900 Sequence Detection System (Applied Biosciences). The expression of target genes was normalised to the housekeeping gene HPRT using the formula ($2^{-\Delta C_t}$). Appendix Table S3 shows the primers used.

### Illumina microarray transcriptome analysis

RNA was extracted from MEFs as described above. The Illumina TotalPrep RNA amplification kit (Ambion) was used to generate cRNA and whole-genome gene expression analysis performed for two independent *Rnaseh2b⁻/⁻ p53⁻/⁻* MEFs and four *Rnaseh2b⁺/⁺ p53⁻/⁻* lines using MouseWG-6 v2.0 Expression BeadChips (Illumina). Microarray data were analysed with R 3.1.0, using the beadarray and Limma v.3.24.15 packages. Raw, non-normalised bead-summary values were imported from the Illumina BeadStudio software into R using the beadarray package. The limma function neqc was used to perform background correction and quantile normalisation of the raw probe signals. Prior to gene level differential analysis, probes that were not detected on any arrays were removed (detection *P*-value < 0.01), and genes with multiple probes were replaced with their average. A linear model was applied to the expression data for each gene. To determine statistically differentially expressed genes, the results of the linear model were summarised and a Bayes moderated *t*-test applied. To control for multiple testing, a Benjamini and Hochberg false discovery rate value of < 0.05 was used. Gene Ontology enrichment analysis was performed using the R package clusterProfiler. Genes in WT MEFs significantly upregulated against *Rnaseh2b* knockout (*q*-value < 0.05 and log2FC > 0) were tested for functional enrichment against the background of genes detected on the array. Microarray data were deposited at the Gene Expression Omnibus (GEO)—accession number GSE76942.

### siRNA knock-down

MEFs were plated overnight at $2 \times 10^4$ cells per well in a 24-well plate. The following day cells were transfected with short-interfering RNA (siRNA) oligonucleotides targeting cGAS, STING or designed against Luciferase, using Dharmafect 1 in Opti-MEM reduced serum medium (Thermo Fisher Scientific). Appendix Table S4 lists the oligonucleotide sequences. Transfection medium was replaced with 800 μl complete medium after 6 h, and culture medium was removed 48 h after transfection and used for ELISA analysis. Total RNA was extracted from cells as described above. To assess responsiveness to poly(I:C) following siRNA treatment, cells were further transfected with a final concentration of 10 μg/ml of high molecular weight poly(I:C) (InvivoGen) and culture supernatants assessed 22 h later for CCL5 using ELISA.

### CRISPR/Cas9 genome editing

*Rnaseh2b⁻/⁻ p53⁻/⁻* MEFs were electroporated using the neon transfection system (Invitrogen) with vectors based on pSpCas9n (BB)-2A-GFP, a gift from Feng Zhang (Addgene plasmid # 48140), expressing two guide RNAs designed against the coding part of exon 1 of murine *cGAS/Mb21d1* (NM_173386.5), using previously described methods (Ran *et al*, 2013). *Rnaseh2b⁻/⁻ p53⁻/⁻ cGAS⁻/⁻* clones were selected by sizing of PCR products of the targeted region to detect clones in which deletions had occurred; these were subsequently confirmed by Sanger sequencing ensuring out of frame deletions in all alleles. *Rnaseh2b⁻/⁻ p53⁻/⁻ cGAS⁺/⁺* MEF clones were identified in parallel to act as controls. An additional functional assay was performed to confirm inactivation of cGAS by assessing cellular responsiveness to transfected dsDNA (1.33 μg/ml final concentration, ISD naked (InvivoGen)) to ensure the absence or presence of functional cGAS, and 2′3′-cGAMP (2 μg/ml final concentration (InvivoGen)) to confirm the presence of functional STING. Innate immune activation of *Rnaseh2b⁻/⁻ p53⁻/⁻ cGAS⁺/⁺* and *Rnaseh2b⁻/⁻ p53⁻/⁻ cGAS⁻/⁻* clones was assessed as described above.

### Complementation of RNase H2 null MEFs

The *Rnaseh2b⁻/⁻ p53⁻/⁻* MEF line used for complementation was previously published and was derived from an E10.5 embryo, resulting from interbreeding *Rnaseh2b^{tm1-hgu-A174T,E202X/+} p53⁺/⁻* mice (Reijns *et al*, 2012). These cells were infected in the presence of 4 μg/ml polybrene with retroviral supernatant from the Phoenix Ecotropic packaging cell line (Swift *et al*, 2001) transfected with pMSCVpuro, allowing expression of EGFP, RNASEH2B (NP_080277) or RNASEH1 (NP_001273794), and selected for stable integration with 2 μg/ml puromycin. Phoenix cells and pMSCVpuro were a kind gift from Dr. Irena Stancheva, University of Edinburgh.

### Immunofluorescence

Parental *Rnaseh2b⁻/⁻ p53⁻/⁻* MEFs, matching *Rnaseh2b⁺/⁺ p53⁻/⁻* controls (Reijns *et al*, 2012) and complemented *Rnaseh2b⁻/⁻ p53⁻/⁻* MEFs ($5 \times 10^4$ cells per well), were plated on coverslips in 6-well plates. After 24 h, non-chromatin bound proteins were extracted using pre-extraction buffer (25 mM HEPES

pH 7.4, 50 mM NaCl, 1 mM EDTA, 3 mM MgCl$_2$, 0.3 M sucrose, and 0.5% Triton X-100) for 7 min on ice. Cells were fixed with 4% PFA for 14 min at room temperature (RT). After blocking with 3% FCS in PBS for 30 min at RT, 53BP1 antibody (NB100-904, Novus Biologicals) was added for 1 h at RT. The next day, Alexa Fluor 568 goat anti-rabbit secondary antibody (Life technologies) was applied and incubated for 1 h at RT. Coverslips were mounted using Vectashield antifade mounting medium with DAPI (Vector laboratories) and imaged at RT using a Coolsnap HQ CCD camera (Photometrics) and a Zeiss Axioplan II fluorescence microscope with Plan-neofluar objectives (×40 and ×63) and acquired with iVision software (BioVision Technologies). Scoring was done under blinded conditions.

### Statistical analysis

Data were analysed by unpaired *t*-test for normally distributed quantitative data and Mann–Whitney *U*-tests for nonparametric data as indicated in the text. One-sample *t*-tests were employed where parametric data were plotted in terms of percentage of control. Prism (GraphPad Software, Inc) was used throughout.

**Expanded View** for this article is available online.

### Acknowledgements

We thank J. Dorin, D. Hunt, J. Rehwinkel, F. Semple and W. Bickmore for helpful discussions and Y. Crow, J. Cambier, J. Rehwinkel, J. Dorin, F. Semple, E. Hall, I. Stancheva and A. Wood for generous sharing of reagents. We thank the IGMM Transgenic Facility, E. Freyer and J. Ding for technical assistance. This work was funded by the MRC, Newlife Foundation for Disabled Children and The Wellcome Trust-University of Edinburgh ISSF2.

### Author contributions

KJM, PC, LL, ZT, RER, BR, FK, AF, PSD, MAMR performed the experiments. KJM, PC, LL, ZT, EA, RER, BR, GG, MAMR and APJ analysed data. FD, AR and AF developed protocols and provided reagents. KJM, KR, RER, MAMR and APJ planned the project/supervised experiments. KJM, MAMR and APJ wrote the manuscript.

### Conflict of interest

The authors declare that they have no conflict of interest.

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
