## [Review Process File · The EMBO Journal]

Manuscript EMBO-2015-93339

Ribonuclease H2 mutations activate the cGAS-STING nucleic-acid sensing pathway

Karen MacKenzie, Paula Carroll, Laura Lettice, Zygimante Tarnauskaite, Kaalak Reddy, Flora Dix, Ailsa Revuelta, Erika Abbondati, Rachel Rigby, Björn Rabe, Fiona Kilanowski, Graeme Grimes, Adeline Fluteau, Paul Devenney, Robert Hill, Martin Reijns and Andrew Jackson

Corresponding author: Andrew Jackson, Institute of Genetics and Molecular Medicine

Review timeline:	Submission date:	23 October 2015
	Editorial Decision:	09 November 2015
	Revision received:	24 December 2015
	Editorial Decision:	14 January 2016
	Revision received:	20 January 2016
	Accepted:	22 January 2016

Editor: Karin Dumstrei

Transaction Report:

1st Editorial Decision 09 November 2015

Thank you for submitting your manuscript to The EMBO Journal. Your study has now been seen by three referees and their comments are provided below.

As you can see, the referees find the analysis interesting and support publication here. They raise a number of constructive points that I would like to ask you to respond to in a revised version. The referee points are clearly outlined below and I will not repeat them here, but do let me know if we need to discuss any of them further. You can use the link below to upload the revised version.

REFeree REPORTS

Referee #1:

In humans, mutations in RNaseH2 can lead to the Aicardi-Goutieres syndrome. The study focuses on the induction of inflammatory cytokine expression in mouse models of RNaseH2 deficiency. They have generated a new mouse model, A174T, and show that it leads to reduced RNaseH activity. They show that expression of select ISGs is increased in heart and kidney, but not in brain. They next switch to the RNaseH2^{-/-}-p53^{-/-} MEFs show that they similarly express more inflammatory cytokines and ISGs. They provide evidences to suggest that cGAS and STING are implicated in both models. Finally, they show that RNaseH2B but not RNaseH1 rescues the DNA damage response and suppression of CCL5 production in RNaseH2B-deficient MEFs.

This is a well-conducted study that provides important insights to the understanding of RNaseH2 activities in mouse models. RNaseH2^{-/-} mice are not viable and an ISG signature was not detected in the aborted embryos. The results presented here with two viable mouse models show an ISG response and are thus important. The significance to human AGS is somewhat limited due to the lack of data on human cells.

Major comments:

- All the experiments were performed in mouse models, which limits the significance to AGS. They have access to human RNASEH2B-A177T cells. Is there increased cytokine and ISG expression in these cells, and if so, can they show this is cGAS/STING mediated?

- In Figure 2: Are the control true genetic littermates? This is not clear from the legend. They need to exclude the possibility that the subtle differences in ISG expression could be due to housing differences, such as subtle differences in microbiota, stress, etc. Is the ISG induction cell-intrinsic? This needs to be addressed: are they able to generate MEFs from RNASEH2B-A174T mice? They should compare ISG induction in these cells to their genetic controls. Finally, CCL5 is missing in this figure, while it is used later as the only marker for ISG signature (Fig. 5) - this is inconsistent.

- In Figure 4A, the siRNA experiment is not well controlled. First, they need to show efficiency and selectivity of knock-down by western blot or qPCR. What is the expression of cGAS and STING in the siRNA treatment for STING and cGAS, respectively (bottom panels 4A)? More importantly, they need to exclude functional off-target effects: for example, they could show that the knock-down cell for cGAS or STING respond to RIG-I or MDA5 agonists similarly to control.

- In Figure 4D, the effect on ISG expression is very subtle; another interpretation is that *Sting*^{-/-} have reduced tonic IFN responses independently of *Rnaseh2b* (see Schoggins, Nature 2014 and Hartlova, Immunity 2015). Is the data on *Rnaseh2b*^{+/+}*sting*^{+/+} and *Rnaseh2b*^{+/+}*sting*^{-/-} littermate controls available? This panel would be the only direct experimental demonstration that STING is required for ISG induction in the A174T/A174T mouse. To corroborate a STING dependent cell intrinsic ISG signature they should show abrogation of ISG induction in primary cells (MEFs) derived from RNaseH2B A174T/A174T STING^{-/-} mice vs their genetic controls.

- In Figure 5, the conclusion is based only on the expression of one cytokine (CCL5) and the magnitude of the rescue by RNaseH2B is low (2-fold; compare 3E and 4B) and not back to WT levels. Thus, the rescue is not convincing. What about the expression of the additional inflammatory cytokines and ISG that are under scrutiny? Is there somehow an impact of the retroviral transduction? Can they show parental mutant cells and WT as comparison?

- The discussion is too limited. They need to discuss why there are differences in ISG induction between their models and the RNaseH2^{-/-}(p53 WT) model. They also need to discuss to what extent the two mouse models do not yet recapitulate AGS. Finally, they should discuss differences between their models and the SAMHD1^{-/-} and TREX1^{-/-} models, in terms of magnitude and specificity of ISG induction and pathogenesis.

Minor comments:

- Figure 4D: Are the mice littermates?

- Figure 3: There is emphasis on differences between MEF lines in the text. Please show individual cell line data for all panels in the figure. What is the explanation for this variability, which seems at odd with the notion that ISG induction would be solely caused by the loss-of-function of H2?

- Figure 4B, 4C: Please show the data supporting the notion that DNA sensing is abrogated in the CRISPR lines and not in control lines.

- There is frequent use of the phrasing "immune response in cells" (i.e. p8), but they have not looked at immune responses, which implicate cells of the immune system and mediate responses to antigens. What they have looked at is not an immune response, but a cell-intrinsic upregulation of a subset of inflammatory cytokines and ISGs.

- "Identification of cGAS", "Identification of the cGAS/STING pathway in RNase H2 deficient cells" (p8) is vague and wrong. Reformulate.

- In the abstract, "We establish Rnaseh2bA174T/A174T knock-in mice as a disease model" is not correct since there is no disease; "The cGAS/STING pathway is therefore the major nucleic acid sensing pathway" is also vague and unsubstantiated. The conclusion (p9) that the findings "implicate the cGAS-STING pathway in RNase H2 AGS, establishing activation of this pathway as the most common cause of AGS" is not true. They have not compared to other pathways, they focused on particular cell types, and they report no pathogenesis. They need to better differentiate conclusions based on direct experimental evidences from speculation.

- End of p3: What is the evidence that cellular RNase H2, or H1, degrades retroviral RNA:DNA hybrids ? This is not described in Cerritelli & Crouch. I don't think that this has been ever demonstrated.

Referee #2:

This is a report on a novel mouse model of Aicardi-Goutières syndrome caused by bi-allelic partial loss-of-function of RNase H2. The authors introduced a point mutation orthologous to the common AGS-associated RNASEH2B variant A177T into the mouse germline. Animals homozygous for this knock in did not develop detectable pathology but show a constitutive activation of innate antiviral immunity. Transcript levels of several type I IFN-inducible genes were upregulated in different tissues of these mice and in Rnaseh2b ko MEFs. Importantly, the IFN response was dependent on a functional cGAS STING axis as demonstrated by siRNA-mediated knock down or CRISPR Cas9-mediated knock out in MEFs. Additional STING deficiency abolished the spontaneous IFN response of the knock in animals in vivo. Spontaneous DNA damage and IFN response of Rnaseh2 ko cells was rescued by reconstitution with RNase H2 but not RNase H1. This is a solid study that concisely reports an important finding.

Minor points:

1. The authors state that the magnitude of ISG induction was similar as reported for SAMHD1^{-/-} mice. I think that this comparison is problematic. A weaker statement would seem more appropriate.
2. It is true that the brain of Trex1^{-/-} mice is spared from inflammation compared to other tissues. However, Pereira-Lopes et al. (J. Immunol 191:6128) demonstrated increased expression of proinflammatory cytokines and ISGs in Trex1 ko brain. This should be mentioned.
3. IFN response but absence of pathology in SAMHD1^{-/-} mice was shown by Rehwinkel et al. but also by Behrendt et al. (Cell Reports 2013). The latter paper should be cited.
4. The authors could have specified the general deleter that was used to excise the loxP-flanked Rnaseh2b exon to yield a null allele.
5. The authors did not delete the neo in their knock in mice. Was this for a particular reason? Can they be sure that the reduced RNaseH2 activity of these animals is entirely due to the point mutation and not in part an effect of altered transcription by the presence of the neo? Was transcription affected?

Referee #3:

Jackson and colleagues make an important contribution to the area of human autoimmune diseases and nucleic acid sensing by innate immunity in this paper, where they develop a Rnaseh2b(A174T/A174T) knock in (KI) mouse. This mutant is linked to the human mono allelic disease AGS. Similar to AGS patients, they find an interferon signature (heightened constitutive

expression of ISGs) in these mice. Using the KI mice, and also Rnaseh2b KO mice, they show that the heightened IFN signature is dependent on cGAS and STING, strongly suggesting that nucleic acids generated due to impaired clearance by RNaseH2b are driving the IFN signature. Previously other mouse models impaired in genes linked to AGS (such as TREX1) have also shown aberrant nucleic acid sensing in a cGAS-STING dependent manner, but the paper here is the first implication of cGAS in immune activation during RNaseH2 deficiency.

Addressing the following issues would further strengthen the paper:

1. Do RNaseH2B KI or KO mice have any phenotypes similar to what is seen in AGS patients with RNaseH2 deficiency? Please comment further on this.
2. Is the ISG expression profile seen in cells from RNASEH2B(A177t/A177T) AGS patients dependent on cGAS and STING - e.g. in the LCL cells (Fig 1E). This would be an important confirmation of the role of cGAS in human RNASEH2B(A177t/A177T) AGS.
3. Figure 2 should show data for CCL5 as well as the other ISGs since CCL5 is focused on in later Figures.
4. The data in Fig 4B and C is critical for the paper in that it shows that complete KO of cGAS by CRISPR/Cas9 completely prevent induction of CCL5 and IFIT1. More ISGs should be measured here too for confirmation. Sometimes downstream ISG induction is specific to certain upstream IFN inducers.
5. Likewise for Fig 5C - this panel is critical in showing RNaseH2b-specific (and not RNaseH1) activity is linked to ISG induction - please measure more ISGs here.
6. It is unfortunate that the authors cannot be more specific about the likely cGAS stimulants generated in the RNaseH2-deficient cells. Can they test for accumulation of dsDNA in the cytosol of these cells? Can they say more about how DNA damage is proposed to generate a cytoplasmic ligand for cGAS?

1st Revision - authors' response

24 December 2015

Response to Referees

We thank the referees for their insightful and supportive comments. We welcome their helpful suggestions which we feel has further strengthened the manuscript. We outline in detail our responses below:

Referee #1:

In humans, mutations in RNaseH2 can lead to the Aicardi-Goutieres syndrome. The study focuses on the induction of inflammatory cytokine expression in mouse models of RNaseH2 deficiency. They have generated a new mouse model, A174T, and show that it leads to reduced RNaseH activity. They show that expression of select ISGs is increased in heart and kidney, but not in brain. They next switch to the RNaseH2-/-p53-/- MEFs show that they similarly express more inflammatory cytokines and ISGs. They provide evidences to suggest that cGAS and STING are implicated in both models. Finally, they show that RNaseH2B but not RNaseH1 rescues the DNA damage response and suppression of CCL5 production in RNaseH2B-deficient MEFs. This is a well-conducted study that provides important insights to the understanding of RNaseH2 activities in mouse models. RNaseH2-/- mice are not viable and an ISG signature was not detected in the aborted embryos. The results presented here with two viable mouse models show an ISG response and are thus important. The significance to human AGS is somewhat limited due to the lack of data on human cells.

We thank the referee for recognising this work as a well-conducted study providing important insights. We appreciate his/her detailed critique, which has helped us further improve the accuracy and informativeness of this manuscript.

Major comments:

1.- All the experiments were performed in mouse models, which limits the significance to AGS. They have access to human RNASEH2B-A177T cells. Is there increased cytokine and ISG expression in these cells, and if so, can they show this is cGAS/STING mediated?

We have not been able to identify a robust ISG signature in the RNASEH2B-A177T patient cell lines available to us. This is likely due to LCLs and primary fibroblasts being suboptimal cell types for these experiments. The ISG response shows tissue to tissue variation in our A174T mouse, and indeed mouse embryonic fibroblasts with this hypomorphic mutation do not have a detectable ISG signature either (see our response to point 3). While an ISG signature is detectable in whole blood from many AGS patients (Rice et al, 2013), this does not necessarily originate from B cells. We are therefore unable to use currently available cell lines to assess whether the ISG response in patients with RNase H2 mutations is cGAS/STING mediated.

It is notable that published work to date investigating innate immune pathways involved in AGS has focussed on mouse models and murine cells (Stetson et al, 2008; Gall et al, 2012; Ablasser et al, 2014; Mannion et al, 2014; Gao et al, 2015; Liddicoat et al, 2015; Pestal et al, 2015), and that the relevance of cGAMP/STING signalling to human disease in AGS has yet to be formally demonstrated. We therefore utilise this valuable point made by Reviewers 1 and 3 in our discussion, to highlight this important outstanding question for the field:

“It will therefore be important to confirm that the innate immune pathways implicated in Trex1, Adar1 and now RNase H2 deficiency through the use of mouse models are also relevant to the human autoinflammatory phenotype in AGS patients in whom these genes are affected.”

2.- In Figure 2: Are the control true genetic littermates? This is not clear from the legend. They need to exclude the possibility that the subtle differences in ISG expression could be due to housing differences, such as subtle differences in microbiota, stress, etc.

The controls are not littermates, and we have revised materials and methods to clarify this. However, we have controlled carefully for genetic background and housing conditions. Our Rnaseh2b-A174T/A174T mice were backcrossed to C57Bl/6J mice to F11, and subsequently maintained as a homozygous mutant line. C57Bl/6J control mice were bought in at 4–6 weeks of age from the same source as those used in backcrossing, to ensure genetic matching to a level of >99.97% identity. All mice were housed in the same facility in the same room in conventional cages, and fed the same water and food until the age of 9 months at which point they were analysed. Furthermore, retrospective assessment of our Sting/RNase H2 inter-cross experiment, specifically comparing littermates, is consistent with Sting dependent ISG induction (please also see our response to the reviewer’s point 8 below).

Most importantly, our reasoning for examining RNase H2 null mouse cells derived entirely independently on a pure C57Bl/6J background, was to exclude subtle environmental or genetic effects as a confounder. The induction of ISGs in this in vitro system was also cGAS/Sting-dependent (Figure 4A-D). We therefore believe that this provides strong additional evidence that the observed ISG induction is independent of environmental factors, such as microbiota, and directly dependent on reduced RNase H2 activity.

In response to the reviewer's question, we now provide detailed documentation of the genetic matching and housing conditions we undertook in materials and methods.

“These were backcrossed to F11 on the C57Bl/6J background to establish congenicity, and subsequently maintained as a homozygous mutant line. C57Bl/6J control mice were bought in at 4-6 weeks of age from the same source as those used in backcrossing, to ensure genetic matching to a level of >99.97% identity. All mice were housed in the same facility in the same room in conventional cages, and fed the same water and food until they were analysed.”

3. Is the ISG induction is cell-intrinsic? This needs to be addressed: are they able to generate MEFs from RNASEH2B-A174T mice? They should compare ISGs induction in these cells to their genetic controls.

We do not detect a significant increase in CCL5 production (see figure below) or ISG expression (data not shown) in *Rnaseh2b*^{A174T/A174T} MEFs. We also examined CCL5 production in a *Rnaseh2b*^{tm1a/tm1a} MEF line (Genetrap, GT) available to us, derived from previously published knockout first mice (Hiller et al, 2012), and compared this to a littermate control *Rnaseh2b*^{+/+} line. The knockout first MEFs retain residual RNase H2 activity and show a small response (see figure below) consistent with a cell intrinsic ISG induction that inversely correlates with level of enzymatic activity.

Homozygous knockout first MEFs (tm1a/tm1a) produce more CCL5 than wildtype littermate control MEFs (+/+), but much less than *Rnaseh2b*^{-/-} *p53*^{-/-} MEFs (as Fig 3 and Appendix Fig S2). Three independent *Rnaseh2b*^{A174T/A174T} MEF lines did not produce more CCL5 than three independent C57Bl/6 control MEF lines (B16 +/+). Each data point represents the mean of three independent experiments for each line.

These findings suggest that MEFs are not as responsive as other cell types, and therefore there may be a threshold of reduced RNase H2 activity that is not reached in the *Rnaseh2b*^{A174T/A174T} MEFs to trigger an ISG response. Such cell-type variation in responsiveness is consistent with the tissue to tissue variation seen in this study (Fig 2) and in previous studies with other AGS mouse models. The ISG response in *Samhd1*^{-/-} mouse cells also varies with cell and tissue-type (Rehwinkel et al, 2013; Behrendt et al, 2013).

In response to the reviewer's comment, we are more cautious in stating the cell intrinsic nature of the ISG response, stating in the discussion only that "The resulting ISG transcriptional response and induction of proinflammatory cytokines are consistent with cell-intrinsic innate immune activation."

4. Finally, CCL5 is missing in this figure, while it is used later as the only marker for ISG signature (Fig. 5) - this is inconsistent.

We now include data for CCL5 in Figure 2.

5.- In Figure 4A, the siRNA experiment is not well controlled. First, they need to show efficiency and selectivity of knock-down by western blot or qPCR. What is the expression of cGAS and STING in the siRNA treatment for STING and cGAS, respectively (bottom panels 4A)? More importantly, they need to exclude functional off-target effects: for example, they could show that the knock-down cell for cGAS or STING respond to RIG-I or MDA5 agonists similarly to control.

As requested, we now include RT-qPCR data showing that cGAS siRNA has no effect on STING transcript levels, and vice versa, STING siRNA on cGAS transcript levels (Fig 4A). This is in addition to our original data showing efficient knock-down of cGAS and STING transcripts by RT-qPCR.

As suggested, we also have performed transfection experiments with high molecular weight poly(I:C), an RLR ligand, and demonstrate that responsiveness to this is retained after STING and cGAS siRNA knockdown, similar to that seen after luciferase siRNA (Appendix Fig S3). Furthermore, the likelihood of off-target effects is also mitigated by siRNA targeting of two distinct components of the pathway, which both give the same outcome. Finally, further independent evidence of the specificity of our findings is provided by the loss of ISG induction 1) *in vitro* using

an entirely different methodology (CRISPR/Cas9 genome editing) to knock out cGAS (Fig 4B-D), and 2) *in vivo* in *Rnaseh2b^{A174T/A174T} Sting^{-/-}* mouse tissue (Fig 4E).

6.- In Figure 4D, the effect on ISG expression is very subtle; another interpretation is that *Sting^{-/-}* have reduced tonic IFN responses independently of *Rnaseh2b* (see Schoggins, *Nature* 2014 and Hartlova, *Immunity* 2015). Is the data on *Rnaseh2b^{+/+}sting^{+/+}* and *Rnaseh2b^{+/+}sting^{-/-}* littermate controls available? This panel would be the only direct experimental demonstration that STING is required for ISG induction in the *A174T/A174T* mouse. To corroborate a STING dependent cell intrinsic ISG signature they should show abrogation of ISGs induction in primary cells (MEFs) derived from *RNaseH2B A174T/A174T STING^{-/-}* mice vs their genetic controls.

We thank the reviewer for pointing out this potential confounding factor to our *in vivo* experiment. While we have not been able to address the reviewer's question in MEFs given point 3 above, we have performed the suggested analysis *in vivo*, assessing ISG transcript levels in cardiac tissue from *Rnaseh2b^{+/+} Sting^{+/+}* and *Rnaseh2b^{+/+} Sting^{-/-}* mice (Fig 4F). Reassuringly, we do not see a reduction in ISG levels in *Sting^{-/-}* hearts compared to those of *Sting^{+/+}* mice, indicating that a tonic difference in IFN does not underlie the reduction in ISG levels in our mice. We therefore conclude that STING is indeed responsible *in vivo* for the observed ISG induction in *Rnaseh2b^{A174T/A174T}* mice.

7.- In Figure 5, the conclusion is based only on the expression of one cytokine (CCL5) and the magnitude of the rescue by *RNaseH2B* is low (2-fold; compare 3E and 4B) and not back to WT levels. Thus, the rescue is not convincing. What about the expression of the additional inflammatory cytokines and ISG that are under scrutiny? Is there somehow an impact of the retroviral transduction? Can they show parental mutant cells and WT as comparison?

We now include qRT-PCR data for IFIT1, IFIT3 and OAS1A transcripts, as well as ELISA data for CXCL10 secretion in Figure 5, and also compare their levels in complemented cells to that in *Rnaseh2b^{+/+}* (wild-type) and *Rnaseh2b^{-/-}* (parental) cells that have not been transduced with retrovirus.

These additional experiments confirm that cells transduced with the *Rnaseh2b* construct return ISG and cytokine expression to wild-type levels. In contrast, complementation with the *Rnaseh1* construct does not. While some reduction is seen with CXCL10 relative to the parental cell line, for other ISGs expression is the same as, if not greater than in uncomplemented cells. Previous retroviral transduction is unlikely to account for altered ISG levels in these stably maintained lines, particularly as cells retrovirally-transduced with an EGFP construct to control for this possibility do not have significantly reduced ISG levels.

- The discussion is too limited. They need to discuss why there are differences in ISG induction between their models and the *RNaseH2^{-/-}(p53 WT)* model. They also need to discuss to what extent the two mouse models do not yet recapitulate AGS. Finally, they should discuss differences between their models and the *SAMHD1^{-/-}* and *TREX1^{-/-}* models, in terms of magnitude and specificity of ISG induction and pathogenesis.

We have revised the discussion to address these points, providing the following text:
 "ISG activation varied between tissues, which may explain why such transcriptional changes were not reported previously in *Rnaseh2b^{-/-}* or *Rnaseh2b^{tm1a/tm1a}* embryos (Hiller et al, 2012; Reijns et al, 2012). The precise factors determining tissue-specific cGAS activation are currently unclear, but could include the sensitivity of the cGAS/STING pathway in particular cell-types, their level of cell proliferation, the rate of accumulation of immunogenic nucleic acids, and the counteracting influences of cellular processes degrading such nucleic acids.

Despite tissue-specific ISG upregulation, *Rnaseh2b^{A174T/A174T}* mice have a subclinical phenotype without any overt inflammatory pathology, and so, like all other AGS mouse models do not recapitulate the neuroinflammation seen in human AGS patients (Rabe, 2013; Behrendt & Roers, 2014). It will therefore be important to confirm that the innate immune pathways implicated in *Trex1*, *Adar1* and now *RNase H2* deficiency through the use of mouse models are also relevant to the human autoinflammatory phenotype in AGS patients in whom these genes are affected. *Samhd1^{-/-}* mice, like *Rnaseh2b^{A174T/A174T}* mice, display an ISG response in the absence of detectable pathology (Behrendt et al, 2013; Rehwinkel et al, 2013), while a strong ISG response in *Adar1* null or editing

deficient mice is associated with embryonic lethality (Mannion et al, 2014; Liddicoat et al, 2015; Pestal et al, 2015), and with autoinflammatory cardiomyopathy and multi-tissue involvement in *Trex1*^{-/-} mice (Morita et al, 2004; Stetson et al, 2008; Gall et al, 2012). The variation in severity between different AGS gene mouse models remains unexplained, although it may be meaningful that mutations in human *RNASEH2B* are associated with the least severe disease course, with AGS onset generally in infancy, in contrast to the prenatal/neonatal onset more commonly seen in *TREX1* patients (Crow et al, 2015). Also, additional triggers, such as viral infection, have been proposed to be relevant to the pathogenesis of AGS (Crow & Manel, 2015). Reports of marked phenotypic variability in sibling pairs with identical *RNASEH2B* or *RNASEH2C* mutations (Vogt et al, 2013; Tüngler et al, 2014) are consistent with the possibility of such environmental factors impacting on disease severity. The *Rnaseh2b*^{A174T/A174T} mouse and other models provide an opportunity for future investigation of these aspects.”

In the results section we also document the different magnitude in response between RNase H2, *Trex1* and *Samhd1* mice.

“The 2 to 4-fold induction of ISGs we observed in *Rnaseh2b*^{A174T/A174T} heart tissue is comparable to the 4 to 7-fold induction seen in *Samhd1*^{-/-} mouse tissue (Rehwinkel et al, 2013). While an overt inflammatory phenotype is seen in *Trex1*^{-/-} mice (Morita et al, 2004; Gall et al, 2012), pathological signs of neuroinflammation are not evident, although ISG upregulation in brain tissue can be detected (Pereira-Lopes et al, 2013).”

Minor comments:

8.- Figure 4D: Are the mice littermates?

No, not all mice used in this experiment were littermates. One of the *Rnaseh2b*^{A174T/A174T} *Sting*^{+/-} mice and three of the *Rnaseh2b*^{A174T/A174T} *Sting*^{-/-} mice were littermates and we highlight these littermates in the graph below. The ISG transcript levels in these mice are consistent with our overall conclusion that the absence of *Sting* reduces the effect of RNase H2 deficiency *in vivo*.

In response to the reviewer’s point, we have provided information in materials and methods to make clear the genetic and environmental matching performed in our experiments (as for point 1 above).

9.- Figure 3: There is emphasis on differences between MEF lines in the text. Please show individual cell line data for all panels in the figure. What is the explanation for this variability, which seems at odd with the notion that ISG induction would be solely caused by the loss-of-function of H2?

As requested, we provide RT-qPCR data (IFIT1, OAS1A, CXCL10 and CCL5) and ELISA data (CXCL10, CCL5) for individual MEF lines in Appendix Figure S2.

ISG induction varies between isogenic, independently derived lines. Such variation has previously also been seen in *Samhd1*^{-/-} MEFs (Behrendt et al, 2013), as well as *in vivo*. Such variability is currently unexplained and could conceivably even be a stochastic phenomenon. Significant phenotypic variability is seen in AGS patients, even in siblings (Vogt et al, 2013; Tüngler et al, 2014), and one possible explanation has been the possible presence of genetic modifiers in individuals from different genetic background. However, our and others’ data in genetically

matched (inbred) mice suggests that this may not be the whole explanation and understanding the reasons for such variability will therefore be an interesting area for future study.

10.- *Figure 4B, 4C: Please show the data supporting the notion that DNA sensing is abrogated in the CRISPR lines and not in control lines.*

We provide this data in Appendix Figure S4. This confirms that the *Rnaseh2b^{-/-} cGas^{-/-}* CRISPR lines (like MEFs derived from *cGas^{-/-}* embryos) no longer produce CCL5 in response to dsDNA, in contrast to *Rnaseh2b^{-/-} cGAS^{+/+}* controls. In addition, we document that both *Rnaseh2b^{-/-} cGAS^{+/+}* and *Rnaseh2b^{-/-} cGAS^{-/-}* MEF lines respond to 2'3'-cGAMP indicating the presence of functional STING signalling.

11- *There is frequent use of the phrasing "immune response in cells" (i.e. p8), but they have not looked at immune responses, which implicate cells of the immune system and mediate responses to antigens. What they have looked at is not an immune response, but a cell-intrinsic upregulation of a subset of inflammatory cytokines and ISGs.*

We thank the reviewer for pointing this out. We do not wish to inadvertently imply that we detected adaptive immune involvement, and therefore revised the text to avoid such phrasing where possible. When we mention immune signalling, we ensure this is with the qualifier 'innate', which we believe is reasonable given that sensing of PAMPs/DAMPs occurs in many tissue types.

12- *"Identification of cGAS", "Identification of the cGAS/STING pathway in RNase H2 deficient cells" (p8) is vague and wrong. Reformulate.*

We agree that this was poorly phrased. We have now changed the relevant sentences to: "Dependence of the ISG response in RNase H2 and Trex1 deficient mouse cells on the cGAS/STING pathway suggests that accumulation of cytoplasmic DNA is common to both TREX1 and RNase H2 AGS, given that dsDNA is the canonical ligand for cGAS (Gao et al, 2013; Sun et al, 2013)."

"Irrespective of the chemical nature of the ligand, our observation that cGAS plays a central role in the ISG response in RNase H2 deficiency provides further impetus for defining the source of the immunogenic nucleic acids."

"This has precluded a definitive answer to date, but may be aided by the identification of cGAS as the nucleic acid sensor that binds immunogenic nucleic acids in both Trex1 and RNase H2 deficient cells, informing potential future biochemical strategies."

13- *In the abstract, "We establish Rnaseh2b^{A174T/A174T} knock-in mice as a disease model" is not correct since there is no disease; "The cGAS/STING pathway is therefore the major nucleic acid sensing pathway" is also vague and unsubstantiated. The conclusion (p9) that the findings "implicate the cGAS-STING pathway in RNase H2 AGS, establishing activation of this pathway as the most common cause of AGS" is not true. They have not compared to other pathways, they focused on particular cell types, and they report no pathogenesis. They need to better differentiate conclusions based on direct experimental evidences from speculation.*

We have made the following changes to the text:

In the abstract we now write "Here, we establish *Rnaseh2b^{A174T/A174T}* knock-in mice as a subclinical model of disease" and "This suggests that the cGAS/STING signalling pathway is a major nucleic acid sensing pathway relevant to AGS, providing additional insight into disease pathogenesis..." More than half of all AGS patients carry RNase H2 mutations, and 23% have TREX1 mutations (Crow et al, 2015). So, if our finding of cGAS/STING-dependence of the ISG response in RNase H2 deficient mouse cells is indeed relevant to the autoinflammation in AGS patients with RNase H2 mutations, and if the same was shown to be true for patients with TREX1 mutations (autoinflammation in *Trex1^{-/-}* mice is cGAS/STING-dependent) this would make it by far the most important pathway. In the discussion we now write "our findings implicate the cGAS-STING pathway in RNase H2 AGS and, together with the previously attributed role in Trex1 deficiency, suggest it is the most common signalling pathway driving inflammation in AGS, relevant also to the pathogenesis of SLE (Lee-Kirsch et al, 2007; Günther et al, 2015)".

14- *End of p3: What is the evidence that cellular RNase H2, or H1, degrades retroviral RNA:DNA*

hybrids? This is not described in Cerritelli & Crouch. I don't think that this has been ever demonstrated.

The referee is correct that no evidence for retroviral RNA:DNA hybrid degradation by cellular RNase H is given in the review by Cerritelli & Crouch (nor does any exist to our knowledge), but these authors simply discuss potential biologically relevant substrates for this class of nucleases, about which surprisingly little is known.

To clarify we have changed this to:

“Although the exact *in vivo* substrates of these nucleases remain to be identified, they are thought to act on RNA:DNA hybrids arising during nuclear DNA replication, R-loop formation and/or (endogenous) retroviral reverse transcription (Cerritelli & Crouch, 2009).”

Referee #2:

This is a report on a novel mouse model of Aicardi-Goutières syndrome caused by bi-allelic partial loss-of-function of RNase H2. The authors introduced a point mutation orthologous to the common AGS-associated RNASEH2B variant A177T into the mouse germline. Animals homozygous for this knock in did not develop detectable pathology but show a constitutive activation of innate antiviral immunity. Transcript levels of several type I IFN-inducible genes were upregulated in different tissues of these mice and in Rnaseh2b ko MEFs. Importantly, the IFN response was dependent on a functional cGAS STING axis as demonstrated by siRNA-mediated knock down or CRISPR Cas9-mediated knock out in MEFs. Additional STING deficiency abolished the spontaneous IFN response of the knock in animals in vivo. Spontaneous DNA damage and IFN response of Rnaseh2 ko cells was rescued by reconstitution with RNase H2 but not RNase H1.

This is a solid study that concisely reports an important finding.

We thank the referee for their enthusiastic and supportive comments.

Minor points:

1. *The authors state that the magnitude of ISG induction was similar as reported for SAMHDI^{-/-} mice. I think that this comparison is problematic. A weaker statement would seem more appropriate.*

We agree. Rehwinkel et al (2013) detected ISG induction in spleens from *Samhd1^{-/-}* mice, with *IFIT2* and *TNF-α* transcript levels raised 7 and 4-fold respectively, but no detectable change in lung, kidney and heart. In hearts from our *Rnaseh2b^{A174T/A174T}* mice we observed a slightly smaller increase in ISG transcript levels: *IFIT1* (2.4-fold), *IFIT3* (2.8-fold), *IFI44* (1.9-fold), *CXCL10* (3.8-fold) and *OAS1A* (1.8-fold). As suggested by the referee, we have therefore revised this sentence to provide more specific information:

“The 2 to 4-fold induction of ISGs we observed in *Rnaseh2b^{A174T/A174T}* heart tissue is comparable to the 4 to 7-fold induction seen in *Samhd1^{-/-}* mouse tissue (Rehwinkel et al, 2013).”

2. *It is true that the brain of Trex1^{-/-} mice is spared from inflammation compared to other tissues. However, Pereira-Lopes et al. (J. Immunol 191:6128) demonstrated increased expression of proinflammatory cytokines and ISGs in Trex1 ko brain. This should be mentioned.*

We thank the reviewer for highlighting this. We have made the appropriate textual changes, and the relevant section now reads as follows:

“While an overt inflammatory phenotype is seen in *Trex1^{-/-}* mice (Morita et al, 2004; Gall et al, 2012), pathological signs of neuroinflammation are not evident, although ISG upregulation in brain tissue can be detected (Pereira-Lopes et al, 2013).”

3. *IFN response but absence of pathology in SAMHDI^{-/-} mice was shown by Rehwinkel et al. but also by Behrendt et al. (Cell Reports 2013). The latter paper should be cited.*

We are grateful to the reviewer for highlighting this unintentional oversight and we now include this reference in our revised manuscript.

4. The authors could have specified the general deleter that was used to excise the loxP-flanked *Rnaseh2b* exon to yield a null allele.

We have now included details in our methods section on the ubiquitous Cre line used in this work: “*Rnaseh2b*^{tm1c/+} offspring were subsequently crossed to Cre745 mice (a kind gift from DJ Kleinjan, University of Edinburgh), containing a CAGGS-Cre construct in which Cre recombinase is under control of a chicken β -actin promoter (Araki et al, 1995; Kleinjan et al, 2006) to excise *Rnaseh2b* exon 5.”

5. The authors did not delete the neo in their knock in mice. Was this for a particular reason? Can they be sure that the reduced RNaseH2 activity of these animals is entirely due to the point mutation and not in part an effect of altered transcription by the presence of the neo? Was transcription affected?

Early-on in our work with these knock-in mice, we also generated a line in which the neo cassette was excised using a general Cre deleter. This decreased enzyme activity to a level similar to what is seen in AGS patient cells homozygous for A177T. However, as we did not detect a clinical phenotype in these mice, we decided to focus on the mice that retained the neo cassette for our further work as they had more pronounced reduction in RNase H2 activity (Fig 1, ~30% of wildtype controls), to increase our chances of observing an inflammatory phenotype.

As the neo cassette does contribute to the lower RNase H2 activity by reducing transcript levels, we have added the following to the main text:

“More pronounced reduction in the mouse cells may be explained by the presence of a Neomycin selection cassette between exon 6 and 7, causing reduced *Rnaseh2b* transcript levels (~60% of wildtype; data not shown).”

Referee #3:

*Jackson and colleagues make an important contribution to the area of human autoimmune diseases and nucleic acid sensing by innate immunity in this paper, where they develop a *Rnaseh2b*(A174T/A174T) knock in (KI) mouse. This mutant is linked to the human mono allelic disease AGS. Similar to AGS patients, they find an interferon signature (heightened constitutive expression of ISGs) in these mice. Using the KI mice, and also *Rnaseh2b* KO mice, they show that the heightened IFN signature is dependent on cGAS and STING, strongly suggesting that nucleic acids generated due to impaired clearance by RNaseH2b are driving the IFN signature. Previously other mouse models impaired in genes linked to AGS (such as TREX1) have also shown aberrant nucleic acid sensing in a cGAS-STING dependent manner, but the paper here is the first implication of cGAS in immune activation during RNaseH2 deficiency.*

We thank the referee for recognising the importance of our work.

Addressing the following issues would further strengthen the paper:

1. Do RNaseH2B KI or KO mice have any phenotypes similar to what is seen in AGS patients with RNaseH2 deficiency? Please comment further on this.

As previously stated in the Results section, we did not detect any histopathological evidence of inflammation in nine A174T/A174T mice (F11 C57Bl/6) when aged to 1 year, nor did we find evidence of increased mortality. This included absence of any evidence of CNS or skin inflammation, distinctive clinical features of AGS; neither did we observe less common complications of AGS, such as cardiomyopathy. The ISG signature we report here therefore represents a sub-clinical rather than overt clinical phenotype in these mice.

In the Results section we now add:

“In particular, there was no evidence of intracranial calcification, leukodystrophy, chilblain vasculitis, or cardiomyopathy, clinical features associated with AGS in humans (Crow et al, 2015).”

2. Is the ISG expression profile seen in cells from RNASEH2B(A177t/A177T) AGS patients

dependent on cGAS and STING - e.g. in the LCL cells (Fig 1E). This would be an important confirmation of the role of cGAS in human RNASEH2B(A177t/A177T) AGS.

We agree. This valuable point was also made by reviewer 1. Unfortunately, as we have not identified a convincing ISG signature in patient cell lines available to us, we are not currently able to address this. Notably, previous studies reporting mouse models of other AGS genes have not extended their findings to AGS patients, perhaps for similar reasons.

We therefore highlight this as an important outstanding question for the field in our discussion “It will therefore be important to confirm that the innate immune pathways implicated in Trex1, Adar1 and now RNase H2 deficiency through the use of mouse models are also relevant to the human autoinflammatory phenotype in AGS patients in whom these genes are affected.”

3. Figure 2 should show data for CCL5 as well as the other ISGs since CCL5 is focused on in later Figures.

We now include data for CCL5 in Figure 2.

4. The data in Fig 4B and C is critical for the paper in that it shows that complete KO of cGAS by CRISPR/Cas9 completely prevent induction of CCL5 and IFIT1. More ISGs should be measured here too for confirmation. Sometimes downstream ISG induction is specific to certain upstream IFN inducers.

In addition to qRT-PCR data for IFIT1 we now provide further data showing significantly downregulated transcript levels of IFIT3, IFI44, OAS1A, and CCL5 (Fig 4D) in addition to reduced CXCL10 secretion from *Rnaseh2b^{-/-} cgas^{-/-}* cells versus *Rnaseh2b^{-/-} cgas^{+/+}* cells (Fig 4C).

5. Likewise for Fig 5C - this panel is critical in showing RNaseH2b-specific (and not RNaseH1) activity is linked to ISG induction - please measure more ISGs here.

As also requested by Reviewer 1, we now include qRT-PCR data for IFIT1, IFIT3 and OAS1A transcripts, as well as ELISA data for CXCL10 secretion in Figure 5.

6. It is unfortunate that the authors cannot be more specific about the likely cGAS stimulants generated in the RNaseH2-deficient cells. Can they test for accumulation of dsDNA in the cytosol of these cells? Can they say more about how DNA damage is proposed to generate a cytoplasmic ligand for cGAS?

The detection and isolation of aberrant nucleic acids in this and other forms of AGS remains a key challenge for the field. This has been a longstanding point of active investigation in our lab, and the detection of RNA:DNA heteroduplexes, single-stranded and double-stranded DNA continue to be pursued. Biochemical isolation of such nucleic acids from the cytoplasm is particularly problematic, given that even trace contamination from nuclei can confound such analysis.

DNA damage has been widely proposed to be a potential source for immunogenic cytoplasmic DNA, and given that impaired RER can cause reduced genome stability, this is a potential source. While previous studies have suggested that DNA fragments resulting from DNA strand breaks accumulate in the cytoplasm, a mechanism for this has not been established.

We expand our discussion to address the reviewer's points.

“So far, we have been unable to ascertain whether cytoplasmic double-stranded nucleic acids accumulate in RNase H2 deficient cells. However, given that impaired RER causes reduced genome stability, DNA fragments resulting from DNA strand breaks are a potential origin of immunogenic cytoplasmic DNA (Ahn et al, 2014; Hartlova et al, 2015; Shen et al, 2015), while an alternative source could be reverse transcribed retroelements, which are known to be activated upon DNA damage (Farkash & Luning Prak, 2006).”

Other revisions to our manuscript:-

Figure S1 has been added to provide further validation by long-range PCR of the successful gene targeting of the A174T knockin allele to the *Rnaseh2b* locus.

References for response to reviewers

- Ablasser A, Hemmerling I, Schmid-Burgk JL, Behrendt R, Roers A, Hornung V (2014) TREX1 deficiency triggers cell-autonomous immunity in a cGAS-dependent manner. *J Immunol* 192: 5993-5997
- Ahn J, Xia T, Konno H, Konno K, Ruiz P, Barber GN (2014) Inflammation-driven carcinogenesis is mediated through STING. *Nat Commun* 5: 5166
- Araki K, Araki M, Miyazaki J, Vassalli P (1995) Site-specific recombination of a transgene in fertilized eggs by transient expression of Cre recombinase. *Proc Natl Acad Sci U S A* 92: 160-164
- Behrendt R, Schumann T, Gerbaulet A, Nguyen LA, Schubert N, Alexopoulou D, Berka U, Lienenklaus S, Peschke K, Gibbert K, Wittmann S, Lindemann D, Weiss S, Dahl A, Naumann R, Dittmer U, Kim B, Mueller W, Gramberg T, Roers A (2013) Mouse SAMHD1 has antiretroviral activity and suppresses a spontaneous cell-intrinsic antiviral response. *Cell Rep* 4: 689-696
- Behrendt R, Roers A (2014) Mouse models for Aicardi-Goutieres syndrome provide clues to the molecular pathogenesis of systemic autoimmunity. *Clin Exp Immunol* 175: 9-16
- Cerritelli SM, Crouch RJ (2009) Ribonuclease H: the enzymes in eukaryotes. *Febs J* 276: 1494-1505
- Crow YJ, Chase DS, Lowenstein Schmidt J, Szykiewicz M, Forte GM, Gornall HL, Oojageer A, Anderson B, Pizzino A, Helman G, Abdel-Hamid MS, Abdel-Salam GM, Ackroyd S, Aeby A, Agosta G, Albin C, Allon-Shalev S, Arellano M, Ariaudo G, Aswani V et al. (2015) Characterization of human disease phenotypes associated with mutations in TREX1, RNASEH2A, RNASEH2B, RNASEH2C, SAMHD1, ADAR, and IFIH1. *Am J Med Genet A* 167a: 296-312
- Crow YJ, Manel N (2015) Aicardi-Goutieres syndrome and the type I interferonopathies. *Nat Rev Immunol* 15: 429-440
- Farkash EA, Luning Prak ET (2006) DNA damage and L1 retrotransposition. *J Biomed Biotechnol* 2006: 37285
- Gall A, Treuting P, Elkon KB, Loo YM, Gale M, Jr., Barber GN, Stetson DB (2012) Autoimmunity initiates in nonhematopoietic cells and progresses via lymphocytes in an interferon-dependent autoimmune disease. *Immunity* 36: 120-131
- Gao D, Li T, Li XD, Chen X, Li QZ, Wight-Carter M, Chen ZJ (2015) Activation of cyclic GMP-AMP synthase by self-DNA causes autoimmune diseases. *Proc Natl Acad Sci U S A* 112: E5699-5705
- Gao P, Ascano M, Wu Y, Barchet W, Gaffney BL, Zillinger T, Serganov AA, Liu Y, Jones RA, Hartmann G, Tuschl T, Patel DJ (2013) Cyclic [G(2',5')pA(3',5')p] is the metazoan second messenger produced by DNA-activated cyclic GMP-AMP synthase. *Cell* 153: 1094-1107
- Günther C, Kind B, Reijns MA, Berndt N, Martinez-Bueno M, Wolf C, Tungler V, Chara O, Lee YA, Hubner N, Bicknell L, Blum S, Krug C, Schmidt F, Kretschmer S, Koss S, Astell KR, Ramantani G, Bauerfeind A, Morris DL et al. (2015) Defective removal of ribonucleotides from DNA promotes systemic autoimmunity. *J Clin Invest* 125: 413-424
- Hartlova A, Erttmann SF, Raffi FA, Schmalz AM, Resch U, Anugula S, Lienenklaus S, Nilsson LM, Kroger A, Nilsson JA, Ek T, Weiss S, Gekara NO (2015) DNA damage primes the type I interferon system via the cytosolic DNA sensor STING to promote anti-microbial innate immunity. *Immunity* 42: 332-343
- Hiller B, Achleitner M, Glage S, Naumann R, Behrendt R, Roers A (2012) Mammalian RNase H2 removes ribonucleotides from DNA to maintain genome integrity. *J Exp Med* 209: 1419-1426
- Kleinjan DA, Seawright A, Mella S, Carr CB, Tyas DA, Simpson TI, Mason JO, Price DJ, van Heyningen V (2006) Long-range downstream enhancers are essential for Pax6 expression. *Dev Biol* 299: 563-581
- Lee-Kirsch MA, Gong M, Chowdhury D, Senenko L, Engel K, Lee YA, de Silva U, Bailey SL, Witte T, Vyse TJ, Kere J, Pfeiffer C, Harvey S, Wong A, Koskenmies S, Hummel O, Rohde K, Schmidt RE, Dominiczak AF, Gahr M et al. (2007) Mutations in the gene encoding the 3'-5' DNA exonuclease TREX1 are associated with systemic lupus erythematosus. *Nat Genet* 39: 1065-1067
- Liddicoat BJ, Piskol R, Chalk AM, Ramaswami G, Higuchi M, Hartner JC, Li JB, Seeburg PH, Walkley CR (2015) RNA editing by ADAR1 prevents MDA5 sensing of endogenous dsRNA as nonself. *Science* 349: 1115-1120
- Mannion NM, Greenwood SM, Young R, Cox S, Brindle J, Read D, Nellaker C, Vesely C, Ponting CP, McLaughlin PJ, Jantsch MF, Dorin J, Adams IR, Scadden AD, Ohman M, Keegan LP, O'Connell MA (2014) The RNA-editing enzyme ADAR1 controls innate immune responses to RNA. *Cell Rep* 9: 1482-1494
- Morita M, Stamp G, Robins P, Dulic A, Rosewell I, Hrivnak G, Daly G, Lindahl T, Barnes DE (2004) Gene-targeted mice lacking the Trex1 (DNase III) 3'5' DNA exonuclease develop inflammatory myocarditis. *Mol Cell Biol* 24: 6719-6727

- Pereira-Lopes S, Celhar T, Sans-Fons G, Serra M, Fairhurst AM, Lloberas J, Celada A (2013) The exonuclease Trex1 restrains macrophage proinflammatory activation. *J Immunol* 191: 6128-6135
- Pestal K, Funk CC, Snyder JM, Price ND, Treuting PM, Stetson DB (2015) Isoforms of RNA-Editing Enzyme ADAR1 Independently Control Nucleic Acid Sensor MDA5-Driven Autoimmunity and Multi-organ Development. *Immunity* 43: 933-944
- Rabe B (2013) Aicardi-Goutieres syndrome: clues from the RNase H2 knock-out mouse. *J Mol Med (Berl)* 91: 1235-1240
- Rehwinkel J, Maelfait J, Bridgeman A, Rigby R, Hayward B, Liberatore RA, Bieniasz PD, Towers GJ, Moita LF, Crow YJ, Bonthron DT, Reis e Sousa C (2013) SAMHD1-dependent retroviral control and escape in mice. *EMBO J* 32: 2454-2462
- Reijns MA, Rabe B, Rigby RE, Mill P, Astell KR, Lettice LA, Boyle S, Leitch A, Keighren M, Kilanowski F, Devenney PS, Sexton D, Grimes G, Holt IJ, Hill RE, Taylor MS, Lawson KA, Dorin JR, Jackson AP (2012) Enzymatic removal of ribonucleotides from DNA is essential for mammalian genome integrity and development. *Cell* 149: 1008-1022
- Rice GI, Forte GM, Szykiewicz M, Chase DS, Aeby A, Abdel-Hamid MS, Ackroyd S, Allcock R, Bailey KM, Balottin U, Barnerias C, Bernard G, Bodemer C, Botella MP, Cereda C, Chandler KE, Dabydeen L, Dale RC, De Laet C, De Goede CG et al. (2013) Assessment of interferon-related biomarkers in Aicardi-Goutieres syndrome associated with mutations in TREX1, RNASEH2A, RNASEH2B, RNASEH2C, SAMHD1, and ADAR: a case-control study. *Lancet Neurol* 12: 1159-1169
- Shen YJ, Le Bert N, Chitre AA, Koo CX, Nga XH, Ho SS, Khatoor M, Tan NY, Ishii KJ, Gasser S (2015) Genome-derived cytosolic DNA mediates type I interferon-dependent rejection of B cell lymphoma cells. *Cell Rep* 11: 460-473
- Stetson DB, Ko JS, Heidmann T, Medzhitov R (2008) Trex1 prevents cell-intrinsic initiation of autoimmunity. *Cell* 134: 587-598
- Sun L, Wu J, Du F, Chen X, Chen ZJ (2013) Cyclic GMP-AMP synthase is a cytosolic DNA sensor that activates the type I interferon pathway. *Science* 339: 786-791
- Tüngler V, Schmidt F, Hieronimus S, Reyes-Velasco C, Lee-Kirsch MA (2014) Phenotypic Variability in a Family with Aicardi-Goutières Syndrome Due to the Common A177T RNASEH2B Mutation. *Case Reports in Clinical Medicine* 3: 153-156
- Vogt J, Agrawal S, Ibrahim Z, Southwood TR, Philip S, Macpherson L, Bhole MV, Crow YJ, Oley C (2013) Striking intrafamilial phenotypic variability in Aicardi-Goutieres syndrome associated with the recurrent Asian founder mutation in RNASEH2C. *Am J Med Genet A* 161a: 338-342

2nd Editorial Decision

14 January 2016

Thank you for submitting your revised manuscript to The EMBO Journal. Your study has now been re-reviewed by referees #1 and 3. As you can see below, both referees appreciate the introduced changes and support publication here.

Referee #1 suggests a few text changes that I would like to ask you to take into consideration in a final revision.

REFEREE REPORTS

Referee #1:

The manuscript has been greatly improved. The comments of the Reviewer were addressed by providing new experimental evidences or by clarifying/correcting the text. Specific remaining comments to be addressed:

1. The response to my question about the implication of RNASEH2 in retroviral reverse transcription is not convincing. They did not provide any reference to support the notion that RNASEH2 might play a role in this. The term "endogenous" and the use of brackets add more confusion. The mention of "(endogenous) retroviral reverse transcription" (p4) as potential RNASEH2 products should be removed from the Introduction section.

2. They have not tested for the role of cGAS in vivo, thus claims such as "The inflammatory response is dependent, both in vitro and in vivo, on the nucleic acid sensor cyclic GMP-AMP synthase (cGAS) and its adaptor STING" (abstract) and "Here, we establish that RNase H2 deficiency leads to cGAS/STING pathway activation in vitro and in vivo."(p9) are not true with regards to cGAS implication in vivo. They need to clearly dissociate their conclusions based on in vivo (STING only) vs in vitro evidences (cGAS and STING) in the abstract and in the text.

3. There is a confusion in the text between activation of an ISG response that depends on cGAS and STING vs. activation of the cGAS and STING pathway. They have not examined the activation of cGAS or STING themselves (e.g. STING localization and phosphorylation, accumulation of 2'3'-cGAMP). This is important because cGAS and STING are known to control the tonic ISG response (Schoggins et al., Nature; Härtlova et al., Immunity), and ISG comprise many known and likely unknown innate sensors. In other words, it may be that cGAS-STING are required for the tonic expression of another innate sensor through IFN, and that it is this other sensor that gets activated in RNASEH2 deficiency.

Hence, the following claims are not proven:

"Ribonuclease H2 mutations activate the cGAS-STING nucleic-acid sensing pathway" (title)

"Hence, cGAS-STING pathway activation also occurs in vivo," (p7)

"Therefore cGAS activation" (p8)

"Here, we establish that RNase H2 deficiency leads to cGAS/STING pathway activation in vitro and in vivo."(p9)

"establishing activation of this pathway as the most common cause of AGS" (p9)

The following phrasing that they use elsewhere is correct:

"Immune activation is dependent on the cGAS-STING nucleic acid sensing pathway" (p6)

They need to carefully rephrase throughout to reflect dependency, not activation.

4. "The cGAS/STING signalling pathway is therefore the major nucleic acid sensing pathway," (abstract) is vague and not necessary. No "nucleic acid sensing" activity is demonstrated in this work. Remove or reformulate, keeping in mind the limits of the experiments presented.

5. The microarray data accession number (GEO) is missing. It is essential to include this number in the publication. They can use the embargo system at GEO to control date of release.

Referee #3:

The authors have addressed most of the referees comments, with either further data and controls, or discussion. As they indicated, there is still a disconnect between 'animal models' or AGS and human patients, and to date they are unable to show an ISG signature in isolated specific cell lines from patients. This is now acknowledged in the Discussion. Nonetheless, the paper represents a significant contribution to our understanding of autoimmunity.

2nd Revision - authors' response

20 January 2016

Response to reviewers' comments: revised manuscript

We thank the reviewers for their further assessment of our paper and their supportive comments. We would like to address the additional points made by Referee #1 as follows:-

Referee #1:

The manuscript has been greatly improved. The comments of the Reviewer were addressed by providing new experimental evidences or by clarifying/correcting the text. Specific remaining comments to be addressed:

1. The response to my question about the implication of RNASEH2 in retroviral reverse transcription is not convincing. They did not provide any reference to support the notion that RNASEH2 might play a role in this. The term "endogenous" and the use of brackets add more

confusion. The mention of "(endogenous) retroviral reverse transcription" (p4) as potential RNASEH2 products should be removed from the Introduction section.

As requested this has now been changed to:-

"Although the exact *in vivo* substrates of these nucleases remain to be identified, they are thought to act on RNA:DNA hybrids such as those arising during nuclear DNA replication and R-loop formation (Cerritelli & Crouch, 2009)"

2. They have not tested for the role of cGAS *in vivo*, thus claims such as "The inflammatory response is dependent, both *in vitro* and *in vivo*, on the nucleic acid sensor cyclic GMP-AMP synthase (cGAS) and its adaptor STING" (abstract) and "Here, we establish that RNase H2 deficiency leads to cGAS/STING pathway activation *in vitro* and *in vivo*." (p9) are not true with regards to cGAS implication *in vivo*. They need to clearly dissociate their conclusions based on *in vivo* (STING only) vs *in vitro* evidences (cGAS and STING) in the abstract and in the text.

In response to the reviewer's comment we have adjusted the text by removing *in vitro* and *in vivo* as follows:-

Abstract:-

"The inflammatory response is dependent on the nucleic acid sensor cyclic GMP-AMP synthase (cGAS) and its adaptor STING, and is associated with reduced cellular ribonucleotide excision repair activity and increased DNA damage."

(p9):-

"Here, we establish that RNase H2 deficiency leads to a proinflammatory response which is dependent upon the cGAS/STING pathway."

3. There is a confusion in the text between activation of an ISG response that depends on cGAS and STING vs. activation of the cGAS and STING pathway. They have not examined the activation of cGAS or STING themselves (e.g. STING localization and phosphorylation, accumulation of 2'3'-cGAMP). This is important because cGAS and STING are known to control the tonic ISG response (Schoggins *et al.*, *Nature*; Härtlova *et al.*, *Immunity*), and ISG comprise many known and likely unknown innate sensors. In other words, it may be that cGAS-STING are required for the tonic expression of another innate sensor through IFN, and that it is this other sensor that gets activated in RNASEH2 deficiency.

Hence, the following claims are not proven:

"Ribonuclease H2 mutations activate the cGAS-STING nucleic-acid sensing pathway" (title)

"Hence, cGAS-STING pathway activation also occurs *in vivo*," (p7)

"Therefore cGAS activation" (p8)

"Here, we establish that RNase H2 deficiency leads to cGAS/STING pathway activation *in vitro* and *in vivo*." (p9)

"establishing activation of this pathway as the most common cause of AGS" (p9)

The following phrasing that they use elsewhere is correct:

"Immune activation is dependent on the cGAS-STING nucleic acid sensing pathway" (p6)

They need to carefully rephrase throughout to reflect dependency, not activation.

Given that we did not detect changes in tonic levels of ISGs in heart tissue from STING^{-/-} mice, the most parsimonious explanation remains direct sensing of nucleic acids by cGAS. However, we take on board the reviewer's point and have modified the text as follows:-

Title: "Ribonuclease H2 mutations induce a cGAS/STING-dependent innate immune response"

p7: "Hence, a STING-dependent ISG response also occurs *in vivo* in *Rnaseh2b*^{A174T/A174T} mice, implicating the cGAS-STING pathway in the ISG induction observed in RNase H2 AGS patients."

p8: "cGAS-dependent ISG induction in *Rnaseh2b*^{-/-} cells is therefore associated with DNA damage and loss of RNase H2-specific activity, rather than an overall reduction in cellular activity against RNA:DNA hybrids."

p9 "Here, we establish that RNase H2 deficiency leads to a proinflammatory response which is dependent upon the cGAS/STING pathway." (NB: This also includes the revision from point 2). The following additional changes have been made in relation to this comment:-

Abstract: "The inflammatory response is dependent on the nucleic acid sensor cyclic GMP-AMP synthase (cGAS) and its adaptor STING, and is associated with reduced cellular ribonucleotide excision repair activity and increased DNA damage."

We have also changed the subheading on p8 to "Loss of RNase H2-specific activity results in ISG induction"

4. *"The cGAS/STING signalling pathway is therefore the major nucleic acid sensing pathway," (abstract) is vague and not necessary. No "nucleic acid sensing" activity is demonstrated in this work. Remove or reformulate, keeping in mind the limits of the experiments presented.*

This has been reformulated:-

"This suggests that cGAS/STING is a key nucleic acid sensing pathway relevant to AGS, providing additional insight into disease pathogenesis relevant to the development of therapeutics for this childhood-onset interferonopathy and adult systemic autoimmune disorders."

5. *The microarray data accession number (GEO) is missing. It is essential to include this number in the publication. They can use the embargo system at GEO to control date of release.*

This has now been done as requested and the GEO data accession number (GSE76942) is now included in the text.

3rd Editorial Decision

22 January 2016

Thanks for sending me the revised manuscript. I have now had chance to take a look at everything and I appreciate the introduced changes. I am therefore very pleased to accept the manuscript for publication here.

Corresponding Author Name: Andrew P. Jackson
 Journal Submitted to: EMBO Journal
 Manuscript Number: EMBOJ-2015-93339